# LICO: Large Language Models for In-Context Molecular Optimization

**Tung Nguyen & Aditya Grover**
Department of Computer Science
University of California, Los Angeles
`{tungnd,adityag}@cs.ucla.edu`

## Abstract

Optimizing black-box functions is a fundamental problem in science and engineering. To solve this problem, many approaches learn a surrogate function that estimates the underlying objective from limited historical evaluations. Large Language Models (LLMs), with their strong pattern-matching capabilities via pretraining on vast amounts of data, stand out as a potential candidate for surrogate modeling. However, directly prompting a pretrained language model to produce predictions is not feasible in many scientific domains due to the scarcity of domain-specific data in the pretraining corpora and the challenges of articulating complex problems in natural language. In this work, we introduce LICO, a general-purpose model that extends arbitrary base LLMs for black-box optimization, with a particular application to the molecular domain. To achieve this, we equip the language model with a separate embedding layer and prediction layer, and train the model to perform in-context predictions on a diverse set of functions defined over the domain. Once trained, LICO can generalize to unseen molecule properties simply via in-context prompting. LICO performs competitively on PMO, a challenging molecular optimization benchmark comprising 23 objective functions, and achieves state-of-the-art performance on its low-budget version PMO-1K.

## 1 Introduction

Black-box optimization (BBO) is the problem of optimizing an unknown, often complex objective function without direct access to its structure or derivatives. This problem is ubiquitous in many science and engineering fields, including material discovery (Hamidieh, 2018), protein engineering (Brookes et al., 2019; Sarkisyan et al., 2016; Angermueller et al., 2020), molecular design (Gaulton et al., 2012), mechanical design (Berkenkamp et al., 2016; Liao et al., 2019), and neural architecture search (Zoph & Le, 2016). Typically, BBO involves an iterative process where each step constructs a surrogate model to approximate the objective function. This model then guides the selection of promising candidates for subsequent evaluation. The main challenge of this approach lies in learning an effective surrogate function that can accurately estimate the objective from limited historical data.

In stark contrast, we have seen impressive generalization abilities of Large Language Models (LLMs) (Brown et al., 2020; Achiam et al., 2023; Bubeck et al., 2023; Team et al., 2023; Touvron et al., 2023a;b; Jiang et al., 2023; 2024) for language-driven reasoning over many kinds of domains. By pretraining on Internet-scale data, LLMs have demonstrated exceptional pattern-matching abilities and generalization from limited observations in both natural language (Brown et al., 2020; Kojima et al., 2022; Wei et al., 2022) and other domains (Lu et al., 2022; Mirchandani et al., 2023; Gruver et al., 2023). This positions LLMs as a promising solution for enhancing surrogate modeling for BBO. Some recent works have indeed shown great potential for using LLMs for solving optimization problems (Yang et al., 2023; Chen et al., 2023; Zhang et al., 2023; Liu et al.). The main idea behind these methods is to frame the optimization problem in natural language, and prompt the language model using previously collected observations to make predictions for new data points (Liu et al.) or to propose better candidates (Yang et al., 2023; Chen et al., 2023; Zhang et al., 2023; Ma et al., 2023; Nie et al., 2023; Meyerson et al., 2023; Lehman et al., 2023; Bradley et al., 2024; Liu et al., 2023a). However, this approach has several limitations. First, performing optimization in the text space requires the problem and solution to be expressed in natural language, thus limiting this approach

to selected domains. Second, the scarcity of domain-relevant data in the text corpora used to train language models poses generalization challenges when using these models for general scientific domains such as molecular optimization. Therefore, existing works have only demonstrated the success of LLMs in neural architecture search (Liu et al.; Chen et al., 2023; Zhang et al., 2023), prompt optimization (Yang et al., 2023), and code generation (Ma et al., 2023; Lehman et al., 2023), corresponding to domains that are well-represented in the training dataset for common language models (Brown et al., 2020; Touvron et al., 2023a; Jiang et al., 2023). Third, relying on verbose textual descriptions for the problem and its solution imposes practical constraints by inflating the context length and reducing the number of historical observations the model can effectively utilize.

In this work, we propose **L**arge **L**anguage Models for **I**n-**C**ontext **O**ptimization (LICO), a general-purpose model that leverages LLMs for black-box optimization, with a particular application to the molecular domain. To generalize a language model to a new scientific domain unseen during pretraining, we equip the model with two embedding layers for embedding the previously collected molecules and their scores, and a prediction head to predict the score of unseen candidates. Intuitively, the embedding layers map the molecules and their scores to the same feature space already learned by the language model, allowing the model to perform in-context learning in this space instead of the raw text space. Unlike previous methods, this approach is applicable to domains that may not be easily described in natural language such as molecular optimization. Moreover, avoiding verbose textual descriptions enables the model to condition on more historical observations, thus scaling better to harder problems that cannot be solved within a few steps.

We train the new layers together with the (frozen) LLM to perform in-context predictions on a family of functions. Specifically, for each function sampled from this family, we condition the model on a set of inputs and their corresponding evaluations, and task the model to predict the function value of the remaining data points. This task mimics surrogate modeling in BBO, where the surrogate model has to iteratively update its estimation of the underlying objective by conditioning on historical data. An ideal function family to train the model should be close to the target objective functions we want to optimize, but also be diverse enough to encourage generalization. Therefore, we propose to combine intrinsic functions and synthetically generated functions for training LICO. Intrinsic functions are inherent properties of the input that are easy to evaluate. In molecular optimization, for example, intrinsic functions include molecular weight, the number of rings, or heavy atom count, which are obtained via simple computation on the molecule. These intrinsic functions are closely related to the actual objective functions we want to optimize such as bioactivities against a target disease. To facilitate generalization outside of the intrinsic functions, we additionally train LICO on synthetic functions defined over the same target domain that are generated by Gaussian Processes. Our empirical evidence shows the importance of learning from both intrinsic and synthetic functions to the performance of the model on downstream tasks. Figure 1 illustrates our approach.

After training, LICO is capable of optimizing a wide range of molecular properties purely via in-context prompting. While the methodology of LICO applies to general scientific domains, in this paper we focus on molecular optimization. This problem plays a pivotal role in advancing drug and material discovery. The complexity of molecular structures and the vastness of the chemical space present unique challenges to black-box optimization algorithms. Moreover, since molecule-relevant data is likely under-represented in the pretraining corpora of existing language models, molecular optimization is a good problem to test the performance and applicability of LICO. We demonstrate the competitive performance of LICO against the leading methods on Practical Molecular Optimization (PMO) (Gao et al., 2022), a challenging molecular optimization benchmark with 23 objective functions. On PMO's low-budget setting, which we term PMO-1M, LICO achieves the best performance and is the highest-ranked method in the benchmark.

## 2 PROBLEM STATEMENT

Let $f : \mathcal{X} \to \mathbb{R}$ be a real-valued function that operates on a $d$-dimensional space $\mathcal{X} \subseteq \mathbb{R}^d$. In black-box optimization (BBO), the goal is to find the input $x^\star$ that maximizes $f$:

$$x^\star \in \arg\max_{x \in \mathcal{X}} f(x), \tag{1}$$

where we do not have direct access to the structure or gradient information of $f$. In molecular optimization, $\mathcal{X}$ is the space of all possible molecules, and $f$ is a certain property of the molecule we

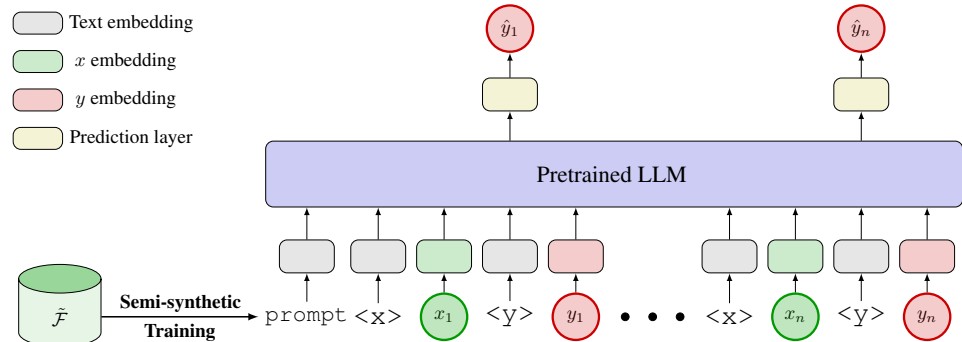

Figure 1: Our proposed approach. We equip a pretrained LLM with an embedding layer for $x$, an embedding layer for $y$, and a prediction layer. We train the model on semi-synthetic data to predict $y$ given $x$ and previous $(x, y)$ pairs. We prepend each $x$ with a special token <x> and each $y$ with a special token <y> to guide in-context reasoning.

want to optimize over, such as bioactivities against a disease. While $f$ is unknown, we often have access to an unlabeled dataset $\mathcal{D}_u$ that consists of molecules $x's$ without the corresponding function values $y's$. ZINC (Sterling & Irwin, 2015) is such a dataset with thousands of unlabeled molecules.

To solve the optimization in equation 1, we can query $f$ with a limited budget, since evaluation often involves expensive physical experiments. To overcome this challenge, a common BBO approach learns a surrogate model $f_\theta$ that approximates the objective $f$ from past observations $\mathcal{D}_{obs} = \{(x_i, y_i)\}_{i=1}^n$, which starts empty and incrementally expands with new data points $(x, f(x))$ we query at each iteration. Formally, a surrogate model represents a predictive distribution $p_\theta(y \mid x, \mathcal{D}_{obs})$ of the function value $y$ conditioned on the input $x$ and the evolving observed dataset $\mathcal{D}_{obs}$. The prediction of this surrogate guides the selection of candidates to balance exploration and exploitation. The newly selected points are added to $\mathcal{D}_{obs}$, and the process continues.

The success of this approach highly depends on the efficiency of the surrogate model $f_\theta$ in estimating $f$ from limited data in $\mathcal{D}_{obs}$ at each iteration. This resembles few-shot prediction, a setting that Large Language Models (LLMs) have proven to excel in. By pretraining on vast Internet-scale data, LLMs can learn generalizable patterns from limited data, and are capable of adapting to multiple functions at test time simply via in-context prompting (Brown et al., 2020; Mirchandani et al., 2023; Krishnamoorthy et al., 2023a;b). A recent line of works (Yang et al., 2023; Zhang et al., 2023; Chen et al., 2023; Liu et al.) has exploited this ability of LLMs for optimization, but they relied on natural language as the interface, thus lacking generality to scientific domains. In this work, we propose a more general and efficient approach to leveraging LLMs for black-box optimization.

## 3 RELATED WORK

**LLMs for Optimization** Recent works have explored the use of LLMs for optimization. The general idea behind these works is to prompt the model with the textual description of the optimization problem and historical evaluations for few-shot reasoning. Yang et al. (2023); Liu et al. (2023a); Zhang et al. (2023); Ma et al. (2023) propose to prompt the language model to directly suggest better candidates to evaluate given the past inputs and their corresponding scores. Meyerson et al. (2023); Lehman et al. (2023); Bradley et al. (2024) integrate LLMs with evolutionary algorithms, and prompt the model to perform crossover and mutation operations based on the population at each optimization step. Liu et al. study the use of LLMs to enhance several components in Bayesian optimization, including warmstarting, surrogate modeling, and candidate generation. Optformer (Chen et al., 2022) proposed to train an LLM specialized for in-context function prediction and optimization.

**LLMs for Molecular Optimization** Recent works have proposed to leverage LLMs for molecular optimization via prompting (Wang et al., 2024; Liu et al., 2023b; Ramos et al., 2023; Völker et al., 2024), leveraging LLM embeddings (Ranković & Schwaller, 2023), or finetuning on molecular corpora (Guevorguian et al., 2024; Ye et al., 2023; Fang et al., 2023; Kristiadi et al.). MOLLEO (Wang et al., 2024) and ChatDrug (Liu et al., 2023b) are two prominent works in the first direction. MOLLEO

proposed to prompt a pretrained LLM to perform crossover and mutation operations in a standard graph genetic algorithm, but its performance largely depends on the prompt format. Similarly, ChatDrug prompts a pretrained LLM for drug editing and requires a retrieval database of molecules to inject domain feedback into the LLM. BoChemian (Ranković & Schwaller, 2023) and Kristiadi et al. studied the use of LLM embeddings for Bayesian molecular optimization, and Kristiadi et al. additionally explored finetuning an LLM to serve as a surrogate for optimization. Chemlactica/Chemma (Guevorguian et al., 2024) and MOLGEN (Fang et al., 2023) proposed to pretrain and/or finetune LLMs on molecule-related corpora to generate valid molecules, which can serve as a genetic algorithm in molecular optimization. DrugAssist (Ye et al., 2023) creates the MolOpt-Instructions dataset that contains pairs of molecules and their property values to finetune a pretrained LLM that can iteratively propose better molecules after training.

The common approach in existing works has several inherent limitations. First, for general scientific domains, the input $x$ may not be easily described by natural language. Second, even when there is a textual description of the input, for instance, molecules can be represented by SMILES strings (Weininger, 1988), existing prompt-based works require significant prompt optimization to achieve good performance, and the optimal prompt often varies between tasks. Furthermore, from an engineering perspective, naively prompting a language model with verbose textual descriptions of the input $x$ results in an excessively long context, thus reducing the number of examples the model can condition on. For example, an LLM with a maximum context length of 4000 can only utilize up to 100 past observations, assuming the average length of each data point is 40. This practically limits the scalability of this approach to harder problems that require more steps to solve.

**LLMs for Non-language Tasks** In addition to optimization, several works have studied the extension of pretrained LLMs to non-language domains with two main directions. The first direction considers problems that can be described in natural language, and prompts a pretrained LLM to solve the problem directly in the text space (Mirchandani et al., 2023; Dinh et al., 2022; Gruver et al., 2023; Liu et al., 2024; Sprueill et al., 2024). The second direction tackles more general problems by learning separate encoders for the new domain and aligning it with the embedding space of the pretrained LLM (Lu et al., 2022; Shen et al., 2023; Tsimpoukelli et al., 2021; Li et al., 2022). Our work is closely related to the latter direction. However, as discussed in the following sections, while many of these works completely leave the word space, we find it beneficial to include language instruction while training the new modules.

## 4 METHOD

We introduce LICO, a methodology for extending arbitrary base LLMs for surrogate modeling in black-box optimization. While the method applies to broad scientific domains, we choose molecular optimization to demonstrate LICO in this paper. We aim to develop a model capable of efficiently adapting to various objective functions after training. To achieve this, we propose a simple extension to existing LLMs and an unsupervised objective using semi-synthetic data to facilitate generalization.

### 4.1 MODEL ARCHITECTURE

In black-box optimization, a surrogate model $f_\theta$ estimates the distribution of the function value $y$ given the input $x$ and past observations $\mathcal{D}_{\text{obs}} = \{(x_i, y_i)\}_{i=1}^n$ the model has collected until the optimization iteration $t$:

$$p_\theta(y \mid x, x_1, y_1, x_2, y_2, \ldots, x_n, y_n), \tag{2}$$

where $x_i$ and $y_i = f(x_i)$ are drawn from an objective function $f$. Our goal is to explore LLMs to model $p_\theta$. As discussed earlier, we make no assumptions on the domain $\mathcal{X}$ to be expressed with natural language. To extend a pretrained language model to an arbitrary new domain, we equip the model with 3 new layers – an embedding layer for the inputs $x's$, an embedding layer for the function values $y's$, and a prediction layer for predicting the unknown function value $y$. Learning separate embedding layers offers several benefits. First, the new embedding layers encode $x$ and $y$ to a shared hidden space obtained by the language model via pretraining, which enables the model to escape the raw text space and perform in-context reasoning in the hidden space instead. Moreover, by embedding each input $x$ to a single hidden vector instead of spanning it over several tokens, we effectively reduce the sequence length and thus allow the model to scale to more conditioning examples.

However, it is challenging for the model to perform this prediction task without any context information about the task. This is because, from the model point of view, embeddings of $x$ and $y$ do not mean anything more than some high-dimensional vectors. In other words, the model does not know what task it should perform and what each token in the embedding sequence represents. To address this issue, we prepend each sequence with a `task prompt` and prepend each input $x$ with a special token `<x>` and each function value $y$ with a special token `<y>`. The `task prompt` instructs the model to perform the task, while the special tokens `<x>` and `<y>` inform the model of the position of each input $x$ and the corresponding function value $y$. In other words, we use a language the model has mastered (natural language) to guide the learning of a new "foreign language" (e.g., molecule). In practice, the `task prompt` is *"Each x is a molecule and each y is the property of the corresponding molecule. Predict y given x."*, whereas `<x>` and `<y>` are two single characters *"x"* and *"y"*. Finally, we apply the prediction layer on top of each token `<y>` to predict the function value given the tokens preceding it. Each prediction consists of a mean and a standard deviation value which will be used for the selection of candidates during optimization. Figure 1 illustrates the architecture of LICO.

It is worth noting that the combination of natural language and domain-specific embeddings is the main distinction between LICO and previous works such as FPT (Lu et al., 2022) which applies pretrained LLMs to *sequence classification* tasks in non-language modalities. FPT also learns new embedding layers for the new domain, but relies entirely on the pretrained self-attention layers to model these embeddings without any language instructions. This distinction stems from the different nature of the tasks we aim to tackle. In sequence classification, the model produces a single prediction for the entire sequence, thus having a good representation of the sequence via self-attention is sufficient. For in-context learning, however, the model must associate each input $x$ with its value $y$ to infer the underlying function $f$ and make predictions for unknown $y$. A language instruction that specifies where $x$ is and where $y$ is helps the model identify this association and improve its in-context reasoning. Our ablation study in 5.2.1 confirms this utility of retaining language tokens.

## 4.2 SEMI-SYNTHETIC TRAINING

Our goal is to train LICO on the unlabeled data $\mathcal{D}_{\mathrm{u}}$ with an unsupervised objective to facilitate efficient generalization to an arbitrary objective function $f$ in the same domain $\mathcal{X}$ after training. Our key insight is that if we train the model to perform the estimation in equation 2 for a wide range of functions, it should adapt to any objective function post-training. While the true function values are unknown before optimization, we can use the unlabeled data $x's$ to generate training data from *other functions*. Assume we have access to a family of functions $\tilde{\mathcal{F}}$ that operate on the same input domain $\mathcal{X}$. For each function $\tilde{f}$ drawn from $\tilde{\mathcal{F}}$, we sample a set of function evaluations $\{(x_i, y_i)\}_{i=1}^n$ and train the model to autoregressively predict $y$ given the input $x$ and preceding $(x, y)$ pairs:

$$\mathcal{L}(\theta) = \mathbb{E}\left[\sum_{i=1}^n \log p_\theta(y_i \mid x_i, x_{<i}, y_{<i})\right], \tag{3}$$

in which the expectation is with respect to $\tilde{f} \sim \tilde{\mathcal{F}}$, $x_{1:n} \sim \mathcal{D}_{\mathrm{u}}$, and $y_{1:n} = \tilde{f}(x_{1:n})$. After training, the estimation in equation 2 can be done purely via in-context prompting, where we condition the model on past observations to make predictions for new data points.

Ideally, the function family $\tilde{\mathcal{F}}$ should be close to the downstream objective $f$, but also be diverse enough to encourage broad generalization across functions. To achieve this, we propose to train LICO on a mix of *intrinsic* and *synthetic* functions, which we term *semi-synthetic* training. Intrinsic functions are functions that map each input molecule $x$ to an inherent property of $x$. For example, molecular weight, the number of rings, or heavy atom count are intrinsic properties of the molecule that are known from domain knowledge or can be easily computed using standard tools. These intrinsic properties are closely related to many downstream objective functions. For example, the biological activity of a drug molecule, such as its ability to inhibit a particular enzyme, is often closely related to the molecule's shape or conformation. Therefore, training LICO from these functions encourages the model to learn useful representations of the input $x$ and obtain good prior knowledge about the optimization domain.

However, it is important to note that we are ultimately interested in optimizing other functions outside of the intrinsic function set. Training the model only on a limited set of intrinsic functions may result in overfitting and poor generalization to unseen functions. To diversify the training data, we

additionally train the model on synthetically generated functions. A synthetic function family should be easy to sample from and be capable of producing diverse functions. Many such families exist, including Gaussian Processes (GPs), randomly constructed Gaussian Mixture Models, or randomly initialized neural networks. We choose to generate synthetic functions from Gaussian Processes with a Tanimoto kernel due to its simplicity and efficiency. Tanimoto kernel, also known as the Jaccard coefficient, measures the similarity between two vectors of binary values, a representation that is widely used for many scientific domains such as chemistry, drug discovery, or bioinformatics. Specifically, each synthetic function $\tilde{f}$ is sampled as follows,

$$\tilde{f} \sim \mathcal{GP}(0, \mathcal{K}), \quad \mathcal{K}(x, x') = \frac{x \cdot x'}{||x||^2 + ||x'||^2 - x \cdot x'}, \tag{4}$$

where $\mathcal{K}(x, x')$ is the Tanimoto kernel that measures the similarity between $x$ and $x'$.

The final family of functions $\tilde{\mathcal{F}}$ used to train LICO is a mixture of intrinsic and synthetic functions with a certain ratio. This design choice is critical to the model's performance. Intuitively, training on both types of functions ensures proximity to the downstream objectives and good coverage of the function space for efficient generalization. The use of intrinsic functions is also the main difference between our work and ExPT (Nguyen et al., 2023), a recent method that studies pure synthetic pretraining for optimization. We hypothesize that while synthetic data is sufficient for ExPT on a few simple tasks, for a more complex domain such as molecular optimization, synthetic training provides too little relevant signal for the model to generalize to downstream objectives. We empirically show the importance of both intrinsic and synthetic functions in the ablation study in section 5.2.2.

## 4.3 LICO FOR BLACK-BOX OPTIMIZATION

After training, a single LICO model can be used for optimizing various objective functions within the domain $\mathcal{X}$. Optimization involves an iterative process. At each iteration $t$, we generate a set of candidates $\{x_i\}_{i=1}^C$ using standard crossover and mutation operations for which the model predicts the mean $\mu_i$ and standard deviation $\sigma_i$ conditioned on prior observations $\mathcal{D}_{\text{obs}}$, a dataset of $(x, y)$ pairs collected until $t$. An acquisition function $\alpha$ then calculates a utility score based on $\mu_i$ and $\sigma_i$ for each candidate, balancing between exploration (favoring high $\sigma$) and exploitation (favoring high $\mu$). The top $k$ candidates determined by their utility scores are evaluated using the objective function $f$. These $k$ candidates and their corresponding evaluations are incorporated into the dataset $\mathcal{D}_{\text{obs}}$, and the cycle repeats. This process terminates once we exhaust the evaluation budget of $B$. Algorithm 1 summarizes the optimization process and Appendix A.3 outlines the optimization hyperparameters.

## 5 EXPERIMENTS

We evaluate LICO on molecular optimization, where the goal is to design new molecules with desired properties such as high chemical stability, low toxicity, or selective inhibition against a target disease. This problem plays a pivotal role in advancing drug and material discovery.

### 5.1 PMO BENCHMARK

**Benchmark** We evaluate LICO on Practical Molecular Optimization (PMO) (Gao et al., 2022), a standard benchmark for molecular optimization with a focus on sample efficiency. We experiment on 23 optimization objectives provided by PMO, including QED Bickerton et al. (2012), DRD2 (Olivecrona et al., 2017), GSK3$\beta$, JNK3 (Li et al., 2018), and 19 objective functions from Guacamol (Brown et al., 2019). QED assesses a molecule's drug-likeness by identifying certain "red flags". DRD2 is a machine learning model trained on experimental data to predict bioactivities for specific target diseases. Guacamol objectives emulate drug discovery goals through a multi-property objective (MPO) approach, considering factors like target molecule similarity, molecular weights, and CLogP. All objective values range from 0 to 1, with 1 indicating the best outcome. We consider two evaluation settings – the original PMO with a budget of 10000 oracle calls, and a budget-efficient setting with 1000 oracle calls, which we refer to as PMO-1K. We believe 1000 is a more reasonable budget while still allowing optimization methods to achieve meaningful performances. To ensure a fair comparison in PMO-1K, we performed extensive hyperparameter tuning for each baseline on the

Table 1: The performance of LICO and the baselines on 23 optimization tasks in PMO-1K. A higher score is better. We report the mean and stddev of scores averaged over 5 random seeds. We use **blue** and **violet** to denote the best and second-best method for each task.

| Task | GP BO | Graph GA | REINVENT | LICO | Genetic GFN | Augmented Memory | MOLLEO |
|---|---|---|---|---|---|---|---|
| albuterol_similarity | $0.636 \pm 0.106$ | $0.583 \pm 0.065$ | $0.496 \pm 0.020$ | $0.656 \pm 0.125$ | $0.664 \pm 0.054$ | $0.557 \pm 0.048$ | $0.886 \pm 0.023$ |
| amlodipine_mpo | $0.519 \pm 0.014$ | $0.501 \pm 0.016$ | $0.472 \pm 0.008$ | $0.541 \pm 0.026$ | $0.534 \pm 0.019$ | $0.489 \pm 0.009$ | $0.637 \pm 0.023$ |
| celecoxib_rediscovery | $0.411 \pm 0.046$ | $0.424 \pm 0.049$ | $0.370 \pm 0.029$ | $0.447 \pm 0.073$ | $0.447 \pm 0.028$ | $0.385 \pm 0.027$ | $0.402 \pm 0.003$ |
| deco_hop | $0.593 \pm 0.013$ | $0.581 \pm 0.006$ | $0.572 \pm 0.006$ | $0.596 \pm 0.010$ | $0.604 \pm 0.017$ | $0.579 \pm 0.010$ | $0.588 \pm 0.007$ |
| drd2 | $0.857 \pm 0.080$ | $0.833 \pm 0.065$ | $0.775 \pm 0.086$ | $0.859 \pm 0.066$ | $0.809 \pm 0.045$ | $0.795 \pm 0.024$ | $0.910 \pm 0.017$ |
| fexofenadine_mpo | $0.707 \pm 0.021$ | $0.666 \pm 0.009$ | $0.650 \pm 0.007$ | $0.700 \pm 0.023$ | $0.682 \pm 0.021$ | $0.679 \pm 0.021$ | $0.674 \pm 0.002$ |
| gsk3b | $0.611 \pm 0.059$ | $0.523 \pm 0.047$ | $0.589 \pm 0.063$ | $0.617 \pm 0.063$ | $0.637 \pm 0.018$ | $0.539 \pm 0.097$ | $0.397 \pm 0.013$ |
| isomers_c7h8n2o2 | $0.545 \pm 0.158$ | $0.735 \pm 0.112$ | $0.725 \pm 0.064$ | $0.779 \pm 0.099$ | $0.738 \pm 0.039$ | $0.661 \pm 0.039$ | $0.737 \pm 0.043$ |
| isomers_c9h10n2o2pf2cl | $0.599 \pm 0.059$ | $0.630 \pm 0.086$ | $0.630 \pm 0.032$ | $0.672 \pm 0.075$ | $0.656 \pm 0.075$ | $0.596 \pm 0.066$ | $0.635 \pm 0.017$ |
| jnk3 | $0.346 \pm 0.067$ | $0.301 \pm 0.071$ | $0.315 \pm 0.042$ | $0.336 \pm 0.051$ | $0.409 \pm 0.165$ | $0.294 \pm 0.110$ | $0.186 \pm 0.076$ |
| median1 | $0.213 \pm 0.020$ | $0.208 \pm 0.015$ | $0.205 \pm 0.012$ | $0.217 \pm 0.019$ | $0.219 \pm 0.008$ | $0.219 \pm 0.014$ | $0.236 \pm 0.021$ |
| median2 | $0.203 \pm 0.009$ | $0.181 \pm 0.009$ | $0.188 \pm 0.010$ | $0.193 \pm 0.009$ | $0.204 \pm 0.011$ | $0.184 \pm 0.010$ | $0.191 \pm 0.009$ |
| mestranol_similarity | $0.427 \pm 0.025$ | $0.362 \pm 0.017$ | $0.379 \pm 0.026$ | $0.423 \pm 0.016$ | $0.414 \pm 0.022$ | $0.393 \pm 0.021$ | $0.399 \pm 0.020$ |
| osimertinib_mpo | $0.766 \pm 0.006$ | $0.751 \pm 0.005$ | $0.737 \pm 0.007$ | $0.759 \pm 0.008$ | $0.763 \pm 0.008$ | $0.761 \pm 0.006$ | $0.779 \pm 0.006$ |
| perindopril_mpo | $0.458 \pm 0.019$ | $0.435 \pm 0.016$ | $0.404 \pm 0.009$ | $0.473 \pm 0.009$ | $0.462 \pm 0.033$ | $0.422 \pm 0.013$ | $0.655 \pm 0.054$ |
| qed | $0.912 \pm 0.010$ | $0.914 \pm 0.007$ | $0.921 \pm 0.002$ | $0.925 \pm 0.005$ | $0.928 \pm 0.002$ | $0.923 \pm 0.002$ | $0.919 \pm 0.006$ |
| ranolazine_mpo | $0.701 \pm 0.023$ | $0.620 \pm 0.014$ | $0.574 \pm 0.044$ | $0.687 \pm 0.029$ | $0.623 \pm 0.022$ | $0.614 \pm 0.033$ | $0.640 \pm 0.000$ |
| scaffold_hop | $0.478 \pm 0.009$ | $0.461 \pm 0.008$ | $0.447 \pm 0.010$ | $0.480 \pm 0.008$ | $0.485 \pm 0.015$ | $0.460 \pm 0.010$ | $0.473 \pm 0.000$ |
| sitagliptin_mpo | $0.232 \pm 0.083$ | $0.229 \pm 0.053$ | $0.261 \pm 0.026$ | $0.315 \pm 0.097$ | $0.227 \pm 0.041$ | $0.245 \pm 0.030$ | $0.193 \pm 0.073$ |
| thiothixene_rediscovery | $0.351 \pm 0.039$ | $0.322 \pm 0.023$ | $0.311 \pm 0.021$ | $0.343 \pm 0.035$ | $0.377 \pm 0.015$ | $0.336 \pm 0.033$ | $0.416 \pm 0.075$ |
| troglitazone_rediscovery | $0.313 \pm 0.018$ | $0.267 \pm 0.015$ | $0.246 \pm 0.009$ | $0.292 \pm 0.028$ | $0.277 \pm 0.015$ | $0.262 \pm 0.012$ | $0.302 \pm 0.022$ |
| valsartan_smarts | $0.000 \pm 0.000$ | $0.000 \pm 0.000$ | $0.000 \pm 0.000$ | $0.000 \pm 0.000$ | $0.000 \pm 0.000$ | $0.000 \pm 0.000$ | $0.000 \pm 0.000$ |
| zaleplon_mpo | $0.392 \pm 0.034$ | $0.374 \pm 0.024$ | $0.406 \pm 0.017$ | $0.404 \pm 0.022$ | $0.400 \pm 0.014$ | $0.415 \pm 0.013$ | $0.392 \pm 0.003$ |
| Sum of scores ($\uparrow$) | 11.27 | 10.90 | 10.68 | **11.71** | 11.56 | 10.81 | **11.65** |

first 5 tasks, and used the optimal hyperparameters for the remaining tasks. Appendix B.1 details hyperparameter search for the baselines.

**Baselines** We compare LICO against 6 leading methods in PMO, namely Genetic GFN (Kim et al., 2024), REINVENT (Olivecrona et al., 2017), Augmented Memory (Guo & Schwaller, 2023), Graph GA (Jensen, 2019), GP BO (Tripp et al., 2021), and MOLLEO (Wang et al., 2024). Genetic GFN employs a GFlowNets (Bengio et al., 2023) model trained to sample molecules proportional to their rewards. REINVENT is a reinforcement learning method that finetunes a pretrained RNN for generating SMILES strings, and Augmented Memory combines REINVENT with data augmentation and experience replay. Graph GA, inspired by evolutionary processes, utilizes crossover and mutation operations to explore the molecule space. GP BO is a Bayesian optimization method that augments Graph GA with a Gaussian Processes surrogate model and UCB acquisition function to guide candidate selection. MOLLEO is an LLM-based method that prompts a chemistry-aware LLM such as BioT5 (Pei et al.) to perform mutation and crossover operations in an evolutionary algorithm. Among the baselines, GP BO is the most similar to LICO, where the only difference is we use an LLM for surrogate modeling instead of a GP.

**LICO training** We use ZINC 250K as the unlabeled dataset $\mathcal{D}_u$. ZINC 250K contains around 250000 molecules sampled from the full ZINC database (Sterling & Irwin, 2015) with moderate size and high pharmaceutical relevance and popularity. We adopt 2-radius 2048 bit Morgan molecular fingerprints as the input feature of the molecule. To generate training data, we use 47 intrinsic properties of the molecule as the intrinsic functions, which we present in detail in Appendix A.1. We train LICO for 20000 iterations with a batch size of 4, where each data point is a sequence of $(x, y)$ pairs sampled from an intrinsic or synthetic function. The ratio of synthetic data is $0.1$. We use Llama-2-7b (Touvron et al., 2023b) as the base LLM, and use LoRA (Hu et al., 2021) for parameter-efficient finetuning. We use the `Llama-2-7b-chat` checkpoint for the 1000-budget setting, and the `Llama-2-7B-32K-Instruct` checkpoint with the Liger Kernel (Hsu et al., 2024) for training and inference with long context for the 10000-budget setting.

**Evaluation details** We report the area under the curve (AUC) of the top-10 average objective value against the number of function calls as the performance metric. AUC metric favors methods that obtain high values with a smaller number of function calls, thus evaluating both optimization capability and sample efficiency. We min-max scale the AUC values to $[0, 1]$. We aggregate the performance for each method across 5 seeds for better reproducibility as suggested by PMO.

**Results** Table 1 summarizes the performance of the 7 considered methods across 23 optimization tasks in PMO-1K. Overall, LICO is the leading method in this benchmark, achieving the highest aggregated score. Specifically, LICO achieves the best or second-best performance in $14/23$ tasks. MOLLEO performs competitively with LICO on this benchmark, with a sum score of $11.65$. However, we note that MOLLEO has significant advantages over LICO and other methods. MOLLEO used

Table 2: The performance of LICO and the baselines on 23 optimization tasks in PMO. A higher score is better. We report the mean and stddev of scores averaged over 5 random seeds. We use **blue** and **violet** to denote the best and second-best method for each task.

| Task | GP BO | Graph GA | REINVENT | LICO | Genetic GFN | Augmented Memory | MOLLEO |
|---|---|---|---|---|---|---|---|
| albuterol_similarity | $0.898 \pm 0.014$ | $0.838 \pm 0.016$ | $0.882 \pm 0.006$ | $0.885 \pm 0.019$ | $0.941 \pm 0.021$ | $0.913 \pm 0.009$ | $0.936 \pm 0.016$ |
| amlodipine_mpo | $0.583 \pm 0.044$ | $0.661 \pm 0.020$ | $0.635 \pm 0.035$ | $0.679 \pm 0.027$ | $0.709 \pm 0.027$ | $0.691 \pm 0.047$ | $0.801 \pm 0.028$ |
| celecoxib_rediscovery | $0.723 \pm 0.053$ | $0.630 \pm 0.097$ | $0.713 \pm 0.067$ | $0.664 \pm 0.122$ | $0.784 \pm 0.032$ | $0.796 \pm 0.008$ | $0.459 \pm 0.080$ |
| deco_hop | $0.629 \pm 0.018$ | $0.619 \pm 0.004$ | $0.666 \pm 0.044$ | $0.619 \pm 0.015$ | $0.653 \pm 0.028$ | $0.658 \pm 0.024$ | $0.648 \pm 0.099$ |
| drd2 | $0.923 \pm 0.017$ | $0.964 \pm 0.012$ | $0.945 \pm 0.007$ | $0.928 \pm 0.018$ | $0.963 \pm 0.006$ | $0.963 \pm 0.006$ | $0.962 \pm 0.013$ |
| fexofenadine_mpo | $0.722 \pm 0.005$ | $0.760 \pm 0.011$ | $0.784 \pm 0.006$ | $0.772 \pm 0.023$ | $0.793 \pm 0.009$ | $0.859 \pm 0.009$ | $0.776 \pm 0.019$ |
| gsk3b | $0.851 \pm 0.041$ | $0.788 \pm 0.070$ | $0.865 \pm 0.043$ | $0.876 \pm 0.045$ | $0.861 \pm 0.022$ | $0.881 \pm 0.021$ | $0.865 \pm 0.037$ |
| isomers_c7h8n2o2 | $0.680 \pm 0.117$ | $0.862 \pm 0.065$ | $0.852 \pm 0.036$ | $0.939 \pm 0.022$ | $0.955 \pm 0.007$ | $0.853 \pm 0.087$ | $0.915 \pm 0.036$ |
| isomers_c9h10n2o2pf2cl | $0.469 \pm 0.180$ | $0.719 \pm 0.047$ | $0.642 \pm 0.054$ | $0.819 \pm 0.039$ | $0.876 \pm 0.018$ | $0.736 \pm 0.051$ | $0.708 \pm 0.093$ |
| jnk3 | $0.564 \pm 0.155$ | $0.553 \pm 0.136$ | $0.783 \pm 0.023$ | $0.731 \pm 0.037$ | $0.759 \pm 0.063$ | $0.739 \pm 0.110$ | $0.715 \pm 0.026$ |
| median1 | $0.301 \pm 0.014$ | $0.294 \pm 0.021$ | $0.356 \pm 0.009$ | $0.291 \pm 0.016$ | $0.355 \pm 0.009$ | $0.326 \pm 0.013$ | $0.302 \pm 0.031$ |
| median2 | $0.297 \pm 0.009$ | $0.273 \pm 0.009$ | $0.276 \pm 0.008$ | $0.280 \pm 0.019$ | $0.289 \pm 0.007$ | $0.291 \pm 0.008$ | $0.206 \pm 0.015$ |
| mestranol_similarity | $0.627 \pm 0.089$ | $0.579 \pm 0.022$ | $0.618 \pm 0.048$ | $0.614 \pm 0.064$ | $0.697 \pm 0.035$ | $0.750 \pm 0.049$ | $0.759 \pm 0.102$ |
| osimertinib_mpo | $0.787 \pm 0.006$ | $0.831 \pm 0.005$ | $0.837 \pm 0.009$ | $0.820 \pm 0.012$ | $0.846 \pm 0.008$ | $0.855 \pm 0.004$ | $0.819 \pm 0.020$ |
| perindopril_mpo | $0.493 \pm 0.011$ | $0.538 \pm 0.009$ | $0.537 \pm 0.016$ | $0.557 \pm 0.028$ | $0.595 \pm 0.011$ | $0.613 \pm 0.015$ | $0.723 \pm 0.018$ |
| qed | $0.937 \pm 0.000$ | $0.940 \pm 0.000$ | $0.941 \pm 0.000$ | $0.936 \pm 0.001$ | $0.937 \pm 0.000$ | $0.942 \pm 0.000$ | $0.933 \pm 0.003$ |
| ranolazine_mpo | $0.735 \pm 0.013$ | $0.728 \pm 0.012$ | $0.760 \pm 0.009$ | $0.774 \pm 0.008$ | $0.810 \pm 0.011$ | $0.801 \pm 0.006$ | $0.731 \pm 0.023$ |
| scaffold_hop | $0.548 \pm 0.019$ | $0.517 \pm 0.007$ | $0.560 \pm 0.019$ | $0.547 \pm 0.026$ | $0.585 \pm 0.041$ | $0.567 \pm 0.008$ | $0.516 \pm 0.022$ |
| sitagliptin_mpo | $0.186 \pm 0.055$ | $0.433 \pm 0.075$ | $0.021 \pm 0.003$ | $0.567 \pm 0.034$ | $0.577 \pm 0.036$ | $0.284 \pm 0.050$ | $0.496 \pm 0.020$ |
| thiothixene_rediscovery | $0.559 \pm 0.027$ | $0.479 \pm 0.025$ | $0.534 \pm 0.013$ | $0.514 \pm 0.037$ | $0.599 \pm 0.073$ | $0.550 \pm 0.041$ | $0.658 \pm 0.024$ |
| troglitazone_rediscovery | $0.410 \pm 0.015$ | $0.390 \pm 0.016$ | $0.441 \pm 0.032$ | $0.380 \pm 0.026$ | $0.455 \pm 0.016$ | $0.540 \pm 0.048$ | $0.352 \pm 0.040$ |
| valsartan_smarts | $0.000 \pm 0.000$ | $0.000 \pm 0.000$ | $0.178 \pm 0.358$ | $0.000 \pm 0.000$ | $0.092 \pm 0.242$ | $0.000 \pm 0.000$ | $0.000 \pm 0.000$ |
| zaleplon_mpo | $0.221 \pm 0.072$ | $0.346 \pm 0.032$ | $0.358 \pm 0.062$ | $0.515 \pm 0.017$ | $0.545 \pm 0.023$ | $0.394 \pm 0.026$ | $0.402 \pm 0.019$ |
| Sum of scores (↑) | 13.156 | 13.751 | 14.196 | 14.708 | 15.678 | 15.002 | 14.682 |

Table 3: Performance of LICO on 5 tasks with different language instructions.

| Task | albuterol_similarity | amlodipine_mpo | celecoxib_rediscovery | deco_hop | drd2 | Sum (↑) |
|---|---|---|---|---|---|---|
| LICO w/o Language | $0.615 \pm 0.104$ | $0.491 \pm 0.018$ | $0.396 \pm 0.051$ | $0.585 \pm 0.010$ | $0.840 \pm 0.063$ | 2.927 |
| LICO w/o Task prompt | $0.641 \pm 0.107$ | $0.523 \pm 0.018$ | $0.457 \pm 0.041$ | $0.595 \pm 0.006$ | $0.844 \pm 0.105$ | 3.060 |
| LICO | $0.656 \pm 0.125$ | $0.541 \pm 0.026$ | $0.447 \pm 0.073$ | $0.596 \pm 0.010$ | $0.859 \pm 0.066$ | 3.099 |

BioT5 to generate valid molecules, which has been finetuned extensively on molecules, protein, and molecule-related text data, while LICO leveraged a general LLM like Llama. Moreover, MOLLEO prompts the LLM with a detailed textual description of the task such as *"Your job is to generate a SELFIES molecule that looks more like a drug"*, which possibly has data contamination issues, since the finetuning data may have included similar tasks. On the other hand, we use LICO as a black-box surrogate model, where the model makes predictions based purely on in-context learning of the mapping between molecule fingerprints and their corresponding scores.

On the original PMO setting, Table 2 shows the competitive performance of LICO with the two state-of-the-art methods Genetic GFN and Augmented Memory. It is important to note that other methods have significant advantages over LICO, since both Genetic GFN and Augmented Memory update their models from real data during optimization, whereas LICO performs in-context learning without being explicitly trained on data from downstream objectives. This impressive result shows the effectiveness of semi-synthetic training in enabling generalization to a broad range of functions via in-context prompting.

The most closely related method to LICO is GP BO, where the only difference between the two is the surrogate model. This indicates the superiority of LICO compared to GP, a popular surrogate model for black-box optimization. To verify this, we compare the predictive performance of LICO and GP on several objective functions. We do this by first labeling the ZINC unlabeled dataset with the objective functions and randomly choosing a subset of the labeled data points for evaluation. For each task, we vary the number of examples given to each method from 32 to 512, and evaluate their performance on 128 held-out data points. We use negative log-likelihood, mean squared error, and root mean squared calibration error as the evaluation metrics. Figure 2 compares the predictive performance of LICO and GP in 3 objective functions, median1, ranolazine_mpo, and troglitazone_rediscovery. The figure shows that the optimization performance of the method closely aligns with the predictive performance of the surrogate model. In median1 and ranolazine_mpo where LICO outperforms GP in terms of optimization score, the model also achieves lower negative log-likelihood, mean squared error, and calibration error. Similarly, LICO has worse predictive performance in troglitazone_rediscovery where it underperforms GP. This verifies our hypothesis and proves the effectiveness of LICO for surrogate modeling.

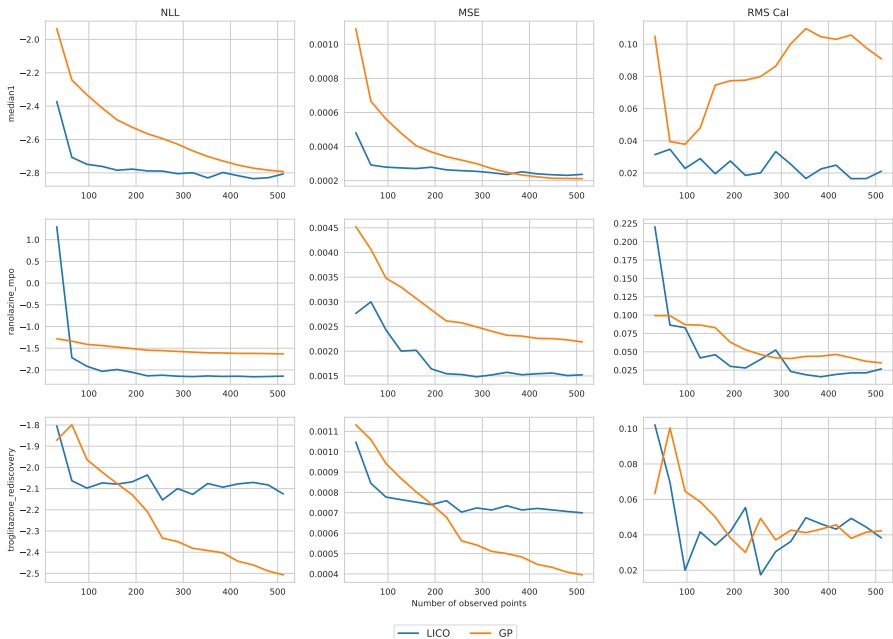

Figure 2: The predictive performance of LICO and GP on 3 objective functions in PMO with different metrics and varying numbers of observations.

Table 4: Performance of LICO on 5 tasks with different ratios of synthetic data.

| Task | albuterol_similarity | amlodipine_mpo | celecoxib_rediscovery | deco_hop | drd2 | Sum (↑) |
|------|---------------------|----------------|----------------------|----------|------|---------|
| LICO Intrinsic | $0.598 \pm 0.115$ | $0.524 \pm 0.029$ | $0.412 \pm 0.042$ | $0.585 \pm 0.005$ | $0.891 \pm 0.032$ | 3.010 |
| LICO 0.1 Synthetic | $0.656 \pm 0.125$ | $\mathbf{0.541 \pm 0.026}$ | $\mathbf{0.447 \pm 0.073}$ | $\mathbf{0.596 \pm 0.010}$ | $0.859 \pm 0.066$ | **3.099** |
| LICO 0.5 Synthetic | $\mathbf{0.663 \pm 0.140}$ | $0.504 \pm 0.016$ | $0.402 \pm 0.016$ | $0.588 \pm 0.006$ | $\mathbf{0.907 \pm 0.020}$ | 3.063 |
| LICO Synthetic | $0.547 \pm 0.080$ | $0.498 \pm 0.026$ | $0.404 \pm 0.103$ | $0.585 \pm 0.003$ | $0.902 \pm 0.012$ | 2.936 |

## 5.2 ABLATION ANALYSIS

We perform various ablation studies to understand the importance of different components and design choices in LICO. For the ablation experiments, we consider the first 5 tasks in Table 1 only. We report the aggregated performance of different models using AUC Top-10 across 5 random seeds.

### 5.2.1 LICO WITHOUT LANGUAGE INSTRUCTION

First, we examine the importance of language instructions to the performance of LICO. We compare 3 variants of LICO: 1) LICO without any language instruction, 2) LICO with special tokens <x> and <y> but without a task prompt, and 3) LICO with both special tokens and the task prompt. Table 3 compares the performance of the 3 variants. LICO performs the best in 4/5 tasks, followed by LICO without the task prompt. LICO without any language instruction performs the worst, often by a large margin. This result confirms the importance of guiding a pretrained LLM with language instruction when applying the model to in-context reasoning in a completely new domain.

### 5.2.2 LICO WITH DIFFERENT SYNTHETIC RATIOS

We investigate the importance of training LICO on both intrinsic and synthetic data. To do this, we gradually increase the ratio of synthetic functions in the training data from 0 (intrinsic-only) to 1 (synthetic-only), and compare the performance of LICO across different ratios. Table 4 shows that LICO with semi-synthetic training performs the best, outperforming both intrinsic-only and synthetic-only data. Training with synthetic data only performs the worst, which is expected when synthetic functions generated by a GP do not include any domain knowledge that is encoded by the intrinsic functions. In other words, synthetic data alone provides too little relevant signal for the model to generalize to unseen downstream objectives. Training with intrinsic functions

Table 5: Performance of pretrained vs randomly initialized LLMs.

| Task | Pretrained LLM | Scratch LLM |
|------|----------------|-------------|
| albuterol_similarity | $\mathbf{0.656 \pm 0.125}$ | $0.575 \pm 0.064$ |
| amlodipine_mpo | $\mathbf{0.541 \pm 0.026}$ | $0.503 \pm 0.029$ |
| celecoxib_rediscovery | $\mathbf{0.447 \pm 0.073}$ | $0.410 \pm 0.034$ |
| deco_hop | $\mathbf{0.596 \pm 0.010}$ | $0.583 \pm 0.005$ |
| drd2 | $\mathbf{0.859 \pm 0.066}$ | $0.827 \pm 0.085$ |
| Sum | $\mathbf{3.099}$ | $2.898$ |

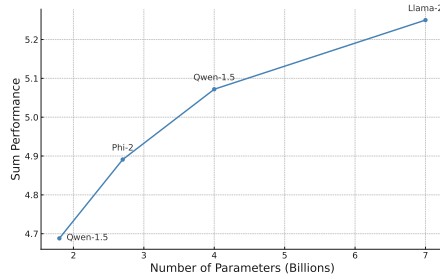

Figure 3: LICO with different LLM sizes.

only, on the other hand, results in quite good performances on most tasks. However, in tasks like `albuterol_similarity`, semi-synthetic training outperforms this baseline by a large margin. We hypothesize that the underlying objective in `albuterol_similarity` is far from the intrinsic functions used to train LICO, leading to poor generalization. Finally, training with small (0.1) to moderate (0.5) ratios of synthetic data achieves similarly good performance.

### 5.2.3 RANDOMLY INITIALIZED VS PRETRAINED LLMS

To understand the importance of using a pretrained LLM, we compare LICO with an autoregressive transformer model of the same size (7B). The transformer architecture is the same as in (Garg et al., 2022), and we train this model to perform in-context learning on the semi-synthetic data from scratch. Table 5 shows the comparison. The scratch model performs much worse than LICO with a pretrained LLM on all tasks despite sharing the same number of parameters. This highlights the importance of the pattern-matching capabilities that LLMs like Llama-2 acquire via extensive language pretraining.

### 5.2.4 LICO WITH DIFFERENT LLM SIZES

Previous works have shown the favorable scaling laws of Large Language Models where larger models consistently perform better on downstream tasks (Kaplan et al., 2020). In this section, we investigate the scaling properties of LLMs but in the context of black-box optimization. Specifically, we compare 4 different base LLMs with different sizes – Qwen-1.5 1.8B and 4B (Bai et al., 2023), Phi-2 2.7B (Javaheripi et al., 2023), and Llama-2 7B (Touvron et al., 2023b). We use the same language instructions for all models. We evaluate each model on the first 8 tasks in Table 1 and average the results across 5 random seeds. We report the sum of performance across 8 tasks.

The comparison in Figure 3 shows that the optimization performance scales consistently with the model size, with Llama-2 7B being the best method. This experiment indicates that larger LLMs not only perform better in language tasks but also obtain stronger pattern-matching capabilities that can be transferred to a completely different domain. Given this scaling, we can further improve the current performance of LICO by scaling up the base LLM size.

## 6 CONCLUSION AND FUTURE WORK

We develop LICO, a new method that leverages pretrained Large Language Models for black-box optimization. LICO extends existing LLMs to non-language domains with separate embedding and prediction layers. To enable efficient generalization to various optimization tasks, we train LICO on a diverse set of semi-synthetic functions for few-shot predictions. LICO achieves state-of-the-art performance on PMO, a challenging molecular optimization benchmark with over 20 objective functions. Ablation analyses highlight the importance of incorporating language instruction to guide in-context learning and semi-synthetic training for better generalization. One limitation of our method is the assumption of an accessible set of intrinsic functions. While this is true for molecular optimization, it may not apply to other scientific domains. In such cases, a better synthetic data generation process incorporating domain knowledge is needed to aid generalization. Future directions include evaluating LICO in other domains to test its applicability and generality, exploring other prompts that better exploit the capabilities of pretrained LLMs, and using LLMs for other aspects of optimization, such as candidate suggestion or exploration.

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

# A  LICO IMPLEMENTATION DETAILS

## A.1  MOLECULAR INTRINSIC FUNCTIONS

We utilize $47$ intrinsic properties of molecules for pretraining LICO. Table 6 shows the intrinsic properties and their explanation.

Table 6: Inherent Properties of Molecules and their Explanations

| Property | Explanation |
| --- | --- |
| Molecular Weight | Total mass of all atoms in the molecule. |
| Number of Rotatable Bonds | Bonds that allow free rotation around themselves. |
| Number of Rings | Count of ring structures in the molecule. |
| Number of H Donors | Atoms in the molecule that can donate a hydrogen atom. |
| Number of H Acceptors | Atoms in the molecule capable of accepting a hydrogen atom. |
| Num Aromatic Rings | Count of rings with a pattern of alternating single and double bonds. |
| Num Aliphatic Rings | Count of non-aromatic rings in the molecule. |
| Num Saturated Rings | Rings with single bonds only. |
| Num Heteroatoms | Atoms other than carbon or hydrogen. |
| Fraction Csp3 | Fraction of carbon atoms bonded with a single pair of electrons. |
| Heavy Atom Count | Count of all atoms except hydrogen. |
| Num Valence Electrons | Total number of electrons that can participate in the formation of chemical bonds. |
| Num Aromatic CarboRings | Aromatic rings composed solely of carbon atoms. |
| Num Aromatic HeteroRings | Aromatic rings containing at least one heteroatom. |
| Num Saturated CarboRings | Saturated rings made only of carbon atoms. |
| Num Saturated HeteroRings | Saturated rings containing at least one heteroatom. |
| BalabanJ | Topological index to quantify molecular branching. |
| BertzCT | A measure of structural complexity of the molecule. |
| Ipc | Information content on the vertex degree. |
| HallKierAlpha | Valence connectivity index used in molecular shape analysis. |
| Kappa1 | Shape descriptor based on the skeleton of the molecule. |
| Kappa2 | Hydrogen suppressed graph descriptor. |
| Kappa3 | Hydrogen complete graph descriptor. |
| Chi0 | Randić molecular connectivity index. |
| Chi1 | Valence modified Randić molecular connectivity index. |
| Chi0n | Randić connectivity index normalized. |
| Chi1n | Valence modified Randić connectivity index normalized. |
| Chi2n | Second order Randić connectivity index normalized. |
| Chi3n | Third order Randić connectivity index normalized. |
| Chi4n | Fourth order Randić connectivity index normalized. |
| Chi0v | Randić connectivity index for valence electrons. |
| Chi1v | First order valence molecular connectivity index. |
| Chi2v | Second order valence molecular connectivity index. |
| Chi3v | Third order valence molecular connectivity index. |
| Chi4v | Fourth order valence molecular connectivity index. |
| Molar Refractivity | Measure of the molecule's polarizability. |
| AMW | Average molecular weight of all atoms in the molecule. |
| Max Partial Charge | Maximum partial charge in the molecule. |
| Min Partial Charge | Minimum partial charge in the molecule. |
| Max Abs Partial Charge | Maximum absolute value of the partial charges in the molecule. |
| Min Abs Partial Charge | Minimum absolute value of the partial charges in the molecule. |
| Labute ASA | Labute's Approximate Surface Area, an estimate of the molecular surface area. |
| Max EState Index | Maximum electrotopological state index of the atoms in the molecule. |
| Min EState Index | Minimum electrotopological state index of the atoms in the molecule. |
| Max Abs EState Index | Maximum absolute value of the electrotopological state indices in the molecule. |
| Min Abs EState Index | Minimum absolute value of the electrotopological state indices in the molecule. |
| fr_C_O | Frequency of carbon-oxygen bonds in the molecule. |

## A.2  TRAINING DETAILS

The $x$ embedding layer, $y$ embedding layer, and prediction layer in LICO are MLPs with a hidden dimension of $1024$. We train LICO for 20000 steps with a batch size of $4$. For each data point in the batch, we randomly decide whether to sample an intrinsic or a synthetic function, with the probability of choosing synthetic functions being $0.1$. Each data point is a sequence of $(x, y)$ pairs with length $n \sim \mathcal{U}[64, 800]$. If the function is an intrinsic function, we uniformly sample a property from Table 6, otherwise sample synthetic data following Equation equation 4.

We use Llama-2-7b (Touvron et al., 2023b) as the base LLM, and use LoRA (Hu et al., 2021) for parameter-efficient finetuning. We use a base learning rate of $5e-4$ with a linear warmup for $1000$

steps and a cosine decay for the remaining 19000 steps. We use LoRA with a rank of 16 and $\alpha$ scale of 16.

### A.3 BLACK-BOX OPTIMIZATION HYPERPARAMETERS

We use Algorithm 1 to optimize a black-box function with LICO. We initialize the observed dataset $\mathcal{D}_{\text{obs}}$ with a population of 34 molecules sampled randomly from ZINC. At each iteration, we use the best 34 candidates in $\mathcal{D}_{\text{obs}}$ to generate new candidates via crossover and mutation operations, with the mutation rate being 0.01. The candidate pool size $C$ is 100. We predict the mean $\mu_i$ and standard deviation $\sigma_i$ for each candidate $x_i$ in the pool using LICO. We employ a UCB acquisition function to compute the utility score $u_i = \mu_i + \beta \sigma_i$, which balances exploration and exploitation. Following (Gao et al., 2022), we set $\beta = 10^b$, where $b \sim \mathcal{U}[-0.5, 1.5]$. We then pick $k = 15$ candidates with the highest utility scores. We evaluate each selected candidate $x_j$ using the black-box function $f$, and add the new data point $(x_j, y_j)$ to the observed dataset $\mathcal{D}_{\text{obs}}$. The process continues with the updated observed dataset, and stops when $|\mathcal{D}_{\text{obs}}| = 1000$.

When predicting $\mu_i, \sigma_i = f_\theta(x_i, \mathcal{D}_{\text{obs}})$, we normalize all the $y's$ values in $\mathcal{D}_{\text{obs}}$ to have mean 0 and standard deviation 1. This is to resemble the finetuning data distribution of LICO. We then denormalize $\mu_i$ and $\sigma_i$ to the original space.

### A.4 BLACK-BOX OPTIMIZATION WITH LICO

Algorithm 1 outlines the optimization algorithm using LICO as the surrogate model.

---

**Algorithm 1** Black-box optimization with LICO

---

**Require:** objective $f$, LICO model $f_\theta$, budget $B$, candidate pool size $C$, acquisition function $\alpha$, batch size $k$
   Initialize $\mathcal{D}_{\text{obs}} = \{\}$
   **while** $|\mathcal{D}_{\text{obs}}| < B$ **do**
      Generate a set of candidates $\{x_i\}_{i=1}^{C}$
      **for** each candidate $x_i$ **do**
         Predict $\mu_i, \sigma_i = f_\theta(x_i, \mathcal{D}_{\text{obs}})$
         Compute utility score $u_i = \alpha(\mu_i, \sigma_i)$
      **end for**
      Select $k$ candidates with the highest utility scores
      **for** each selected candidate $x_j$ **do**
         Evaluate $x_j$ using the actual objective $y_j = f(x_j)$
         Add $(x_j, y_j)$ to the observation dataset $\mathcal{D}_{\text{obs}}$
      **end for**
   **end while**

---

## B BASELINE DETAILS

**TNP** is a transformer-based architecture for in-context learning. We refer to Nguyen & Grover (2022) for more details about TNP. We train a TNP model with 16 attention layers and 2048 hidden dimensions. Other hyperparameters are the same as for LICO. After training, we use TNP for black-box optimization using Algorithm 1 with the same optimization hyperparameters but replace LICO with TNP.

**GP BO** replaces the LICO surrogate model in Algorithm 1 with a Gaussian Process with a Tanimoto kernel. We optimize the Gaussian Process hyperparameters via maximum likelihood estimation on the initial population sampled from ZINC.

**Graph GA** is a model-free variant of Algorithm 1. Specifically, at each iteration, Graph GA generates a set of candidates using the same crossover and mutation operations, and directly evaluates and adds them to $\mathcal{D}_{\text{obs}}$, since it does not employ a surrogate model.

**REINVENT** adopts a policy-based RL approach to finetune a pretrained RNN to generate SMILES strings with high returns. At each optimization iteration, we sample a set of molecules from the

finetuned RNN, evaluate these molecules using the black-box function $f$, and add the new data points to $\mathcal{D}_{\text{obs}}$. We refer to Gao et al. (2022) for more details of the algorithm and other hyperparameters.

## B.1 HYPERPARAMETER TUNING FOR THE BASELINES

To ensure the baselines achieve the best possible performance for the new PMO-1K benchmark, we performed extensive hyperparameter tuning for each baseline on the first 5 tasks with grid search, and used the optimal hyperparameters for the rest of the tasks. Table 7 specifies the grid search for each method.

Table 7: Grid search and optimal hyperparameters for the baseline methods.

| Method | Grid Search | Optimal Hyperparameters |
|---|---|---|
| **GP BO** | `population_size ∈ {50, 100, 150, ..., 350}`
`offspring_size ∈ {50, 100, 150}`
`kept_offspring_size ∈ {5, 10, ..., 50}` | `population_size = 50`
`offspring_size = 100`
`kept_offspring_size = 15` |
| **Graph GA** | `population_size ∈ {50, 100, 150, ..., 350}`
`offspring_size ∈ {50, 100, 150}` | `population_size = 50`
`offspring_size = 100` |
| **REINVENT** | `batch_size ∈ {4, 8, 16, 32, 64}`
`experience_replay ∈ {4, 8, 16, 24, 32}` | `batch_size = 16`
`experience_replay = 24` |
| **Genetic GFN** | `learning_rate ∈ {0.0001, 0.0005}`
`batch_size ∈ {4, 8, 16, 32, 64}`
`num_keep ∈ {128, 256, 512, 1024}`
`offspring_size ∈ {2, 4, 8}`
`ga_generations ∈ {1, 2}` | `learning_rate = 0.0001`
`batch_size = 8`
`num_keep = 128`
`offspring_size = 2`
`ga_generations = 1` |
| **Augmented Memory** | `batch_size ∈ {4, 8, 16, 32, 64}`
`replay_buffer_size ∈ {50, 100, 150}` | `batch_size = 32`
`replay_buffer_size = 100` |

## C ADDITIONAL RESULTS

### C.1 ADDITIONAL METRICS

In addition to AUC Average Top-10, we measure the optimization performance of different methods on AUC Average Top-1 and AUC Average Top-100 for a more comprehensive comparison. Table 8 and 9 show AUC Average Top-1 and AUC Average Top-100 performances, respectively.

Table 8: The performance of LICO and the baselines on 21 optimization tasks in PMO with AUC Average Top-1 metric. A higher score is better. We report the mean and stddev of scores averaged over 5 random seeds. We use **blue** and **violet** to denote the best and second-best method for each task.

| Task | GP BO | Graph GA | LICO | REINVENT | TNP |
|---|---|---|---|---|---|
| albuterol_similarity | $0.672 \pm 0.109$ | $0.647 \pm 0.080$ | $0.695 \pm 0.150$ | $0.572 \pm 0.026$ | $0.611 \pm 0.042$ |
| amlodipine_mpo | $0.538 \pm 0.016$ | $0.526 \pm 0.017$ | $0.560 \pm 0.026$ | $0.500 \pm 0.016$ | $0.513 \pm 0.016$ |
| celecoxib_rediscovery | $0.434 \pm 0.052$ | $0.466 \pm 0.062$ | $0.492 \pm 0.079$ | $0.415 \pm 0.031$ | $0.482 \pm 0.067$ |
| deco_hop | $0.598 \pm 0.013$ | $0.590 \pm 0.005$ | $0.603 \pm 0.012$ | $0.585 \pm 0.010$ | $0.597 \pm 0.002$ |
| drd2_current | $0.895 \pm 0.067$ | $0.898 \pm 0.048$ | $0.902 \pm 0.055$ | $0.867 \pm 0.077$ | $0.831 \pm 0.043$ |
| fexofenadine_mpo | $0.728 \pm 0.022$ | $0.691 \pm 0.011$ | $0.719 \pm 0.025$ | $0.696 \pm 0.012$ | $0.706 \pm 0.014$ |
| isomers_c7h8n2o2 | $0.576 \pm 0.154$ | $0.815 \pm 0.120$ | $0.834 \pm 0.109$ | $0.846 \pm 0.070$ | $0.761 \pm 0.145$ |
| isomers_c9h10n2o2pf2cl | $0.644 \pm 0.053$ | $0.708 \pm 0.083$ | $0.714 \pm 0.084$ | $0.724 \pm 0.043$ | $0.701 \pm 0.086$ |
| median1 | $0.235 \pm 0.016$ | $0.233 \pm 0.018$ | $0.242 \pm 0.020$ | $0.229 \pm 0.015$ | $0.238 \pm 0.015$ |
| median2 | $0.212 \pm 0.010$ | $0.193 \pm 0.011$ | $0.201 \pm 0.009$ | $0.209 \pm 0.013$ | $0.200 \pm 0.018$ |
| mestranol_similarity | $0.449 \pm 0.028$ | $0.387 \pm 0.020$ | $0.445 \pm 0.014$ | $0.433 \pm 0.034$ | $0.406 \pm 0.011$ |
| osimertinib_mpo | $0.788 \pm 0.008$ | $0.777 \pm 0.008$ | $0.781 \pm 0.007$ | $0.780 \pm 0.009$ | $0.776 \pm 0.007$ |
| perindopril_mpo | $0.475 \pm 0.019$ | $0.460 \pm 0.025$ | $0.492 \pm 0.011$ | $0.432 \pm 0.010$ | $0.457 \pm 0.012$ |
| qed | $0.926 \pm 0.011$ | $0.930 \pm 0.004$ | $0.935 \pm 0.002$ | $0.934 \pm 0.003$ | $0.931 \pm 0.001$ |
| ranolazine_mpo | $0.729 \pm 0.024$ | $0.684 \pm 0.015$ | $0.711 \pm 0.028$ | $0.657 \pm 0.048$ | $0.669 \pm 0.032$ |
| scaffold_hop | $0.486 \pm 0.010$ | $0.475 \pm 0.008$ | $0.491 \pm 0.013$ | $0.468 \pm 0.010$ | $0.484 \pm 0.019$ |
| sitagliptin_mpo | $0.268 \pm 0.098$ | $0.281 \pm 0.069$ | $0.363 \pm 0.114$ | $0.333 \pm 0.030$ | $0.274 \pm 0.044$ |
| thiothixene_rediscovery | $0.371 \pm 0.046$ | $0.351 \pm 0.029$ | $0.368 \pm 0.041$ | $0.345 \pm 0.026$ | $0.332 \pm 0.041$ |
| troglitazone_rediscovery | $0.329 \pm 0.019$ | $0.289 \pm 0.021$ | $0.309 \pm 0.033$ | $0.276 \pm 0.009$ | $0.286 \pm 0.012$ |
| valsartan_smarts | $0.000 \pm 0.000$ | $0.000 \pm 0.000$ | $0.000 \pm 0.000$ | $0.000 \pm 0.000$ | $0.000 \pm 0.000$ |
| zaleplon_mpo | $0.431 \pm 0.031$ | $0.418 \pm 0.022$ | $0.435 \pm 0.027$ | $0.456 \pm 0.020$ | $0.428 \pm 0.022$ |
| Sum of scores (↑) | 10.784 | 10.818 | 11.291 | 10.755 | 10.683 |
| Mean rank (↓) | 2.52 | 3.57 | 1.62 | 3.48 | 3.75 |

Table 9: The performance of LICO and the baselines on 21 optimization tasks in PMO with AUC Average Top-100 metric. A higher score is better. We report the mean and stddev of scores averaged over 5 random seeds. We use **blue** and **violet** to denote the best and second-best method for each task.

| Task | GP BO | Graph GA | LICO | REINVENT | TNP |
|---|---|---|---|---|---|
| albuterol_similarity | $0.548 \pm 0.100$ | $0.470 \pm 0.042$ | $0.563 \pm 0.093$ | $0.395 \pm 0.012$ | $0.448 \pm 0.028$ |
| amlodipine_mpo | $0.458 \pm 0.008$ | $0.422 \pm 0.014$ | $0.486 \pm 0.025$ | $0.407 \pm 0.005$ | $0.420 \pm 0.013$ |
| celecoxib_rediscovery | $0.363 \pm 0.040$ | $0.346 \pm 0.036$ | $0.372 \pm 0.070$ | $0.296 \pm 0.024$ | $0.346 \pm 0.026$ |
| deco_hop | $0.579 \pm 0.013$ | $0.563 \pm 0.006$ | $0.583 \pm 0.009$ | $0.550 \pm 0.006$ | $0.568 \pm 0.004$ |
| drd2_current | $0.741 \pm 0.097$ | $0.605 \pm 0.086$ | $0.725 \pm 0.092$ | $0.615 \pm 0.098$ | $0.556 \pm 0.095$ |
| fexofenadine_mpo | $0.645 \pm 0.018$ | $0.588 \pm 0.008$ | $0.636 \pm 0.022$ | $0.549 \pm 0.004$ | $0.599 \pm 0.016$ |
| isomers_c7h8n2o2 | $0.300 \pm 0.142$ | $0.535 \pm 0.091$ | $0.450 \pm 0.149$ | $0.511 \pm 0.058$ | $0.492 \pm 0.115$ |
| isomers_c9h10n2o2pf2cl | $0.474 \pm 0.038$ | $0.441 \pm 0.068$ | $0.535 \pm 0.067$ | $0.445 \pm 0.027$ | $0.447 \pm 0.049$ |
| median1 | $0.175 \pm 0.022$ | $0.168 \pm 0.013$ | $0.166 \pm 0.019$ | $0.162 \pm 0.007$ | $0.170 \pm 0.008$ |
| median2 | $0.184 \pm 0.006$ | $0.158 \pm 0.006$ | $0.175 \pm 0.010$ | $0.155 \pm 0.006$ | $0.162 \pm 0.009$ |
| mestranol_similarity | $0.379 \pm 0.020$ | $0.311 \pm 0.016$ | $0.361 \pm 0.030$ | $0.302 \pm 0.016$ | $0.314 \pm 0.003$ |
| osimertinib_mpo | $0.706 \pm 0.006$ | $0.667 \pm 0.008$ | $0.694 \pm 0.010$ | $0.623 \pm 0.014$ | $0.671 \pm 0.006$ |
| perindopril_mpo | $0.405 \pm 0.019$ | $0.357 \pm 0.012$ | $0.424 \pm 0.007$ | $0.332 \pm 0.011$ | $0.359 \pm 0.010$ |
| qed | $0.853 \pm 0.010$ | $0.854 \pm 0.011$ | $0.882 \pm 0.007$ | $0.874 \pm 0.003$ | $0.857 \pm 0.003$ |
| ranolazine_mpo | $0.633 \pm 0.020$ | $0.462 \pm 0.022$ | $0.617 \pm 0.021$ | $0.436 \pm 0.040$ | $0.468 \pm 0.042$ |
| scaffold_hop | $0.462 \pm 0.006$ | $0.435 \pm 0.008$ | $0.462 \pm 0.006$ | $0.415 \pm 0.009$ | $0.440 \pm 0.010$ |
| sitagliptin_mpo | $0.133 \pm 0.062$ | $0.103 \pm 0.032$ | $0.171 \pm 0.045$ | $0.134 \pm 0.016$ | $0.100 \pm 0.023$ |
| thiothixene_rediscovery | $0.311 \pm 0.030$ | $0.270 \pm 0.015$ | $0.299 \pm 0.026$ | $0.256 \pm 0.015$ | $0.261 \pm 0.024$ |
| troglitazone_rediscovery | $0.283 \pm 0.014$ | $0.228 \pm 0.008$ | $0.258 \pm 0.024$ | $0.201 \pm 0.008$ | $0.230 \pm 0.005$ |
| valsartan_smarts | $0.000 \pm 0.000$ | $0.000 \pm 0.000$ | $0.000 \pm 0.000$ | $0.000 \pm 0.000$ | $0.000 \pm 0.000$ |
| zaleplon_mpo | $0.301 \pm 0.036$ | $0.258 \pm 0.016$ | $0.318 \pm 0.018$ | $0.296 \pm 0.009$ | $0.257 \pm 0.013$ |
| Sum of scores (↑) | 8.933 | 8.242 | 9.175 | 7.954 | 8.167 |
| Mean rank (↓) | 1.95 | 3.62 | 1.76 | 4.14 | 3.45 |

## C.2 LICO WITH LLMS TRAINED ON MOLECULAR CORPORA

One may wonder whether using an LLM finetuned on molecular corpora helps improve the performance of LICO. To answer this question, we compare the performance of LICO with two different base LLMs: T5-base (Raffel et al., 2020) and Nach0-base (Livne et al., 2024), which finetunes T5-base on molecule corpora. Due to time constraints, we did not perform optimization with these new models, but compared their predictive performance instead. Figure 4 summarizes the results. There are two interesting observations from this figure. First, Nach0 works significantly better than plain T5, confirming the hypothesis that proper finetuning of a language model on molecule data helps boost its in-context property prediction in LICO. Second, Llama-2 works better than Nach0. As we explained in the paper, because we perform in-context learning in the embedding space of the language model, we rely on the general pattern-matching capability of the model, i.e., the ability to extract the relationship between embeddings of x and y from examples. From this perspective, it is not surprising that Llama-2 works better than a T5-based model, since it is much larger and has been pretrained with a lot more data, leading to a superior pattern-matching capability. One more benefit of using general LLMs like Llama is that they are domain-agnostic, which means we can finetune them for other non-molecule domains as well.

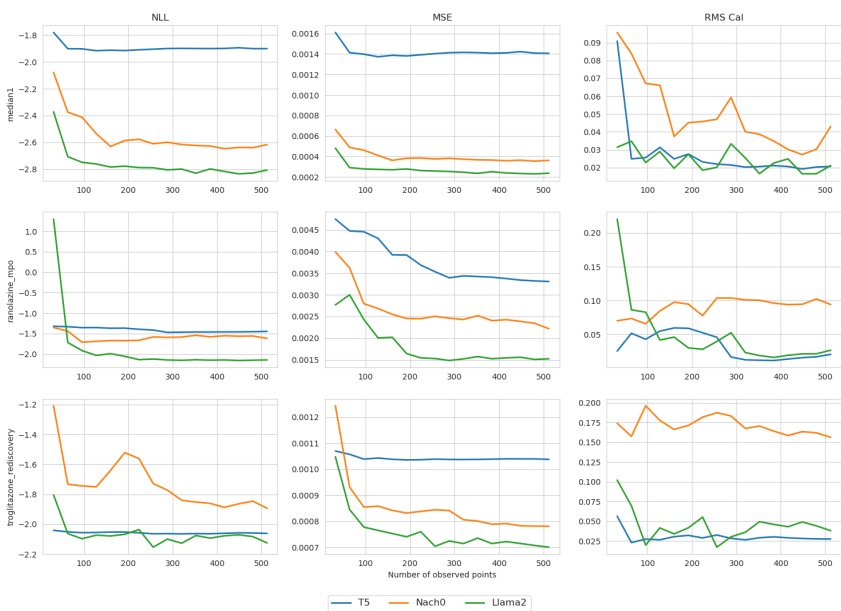

Figure 4: Predictive performance of LICO with T5-base, Nach0, and Llama-2 as the backbones.

## C.3 LICO VS GPT-4 FOR MOLECULAR PROPERTY PREDICTION

One may wonder if we can prompt state-of-the-art LLMs like GPT-4 to perform molecular property prediction. To investigate this, we conducted an experiment where we prompted GPT-4o to perform in-context property prediction in the text space. The following text box shows the prompt we used.

> I will give you a list of molecules and their corresponding property values. Based on these examples, your task is to predict the property value of a new molecule. Please provide your answer as a single number placed inside a pair of parentheses without any other information. For example, if you think the property value of the new molecule is 0.5, you should write 0.5.
> Molecule: $[m_1]$, Property: $[p_1]$
> Molecule: $[m_2]$, Property: $[p_2]$
> ...
> Molecule: $[m_n]$, Property:

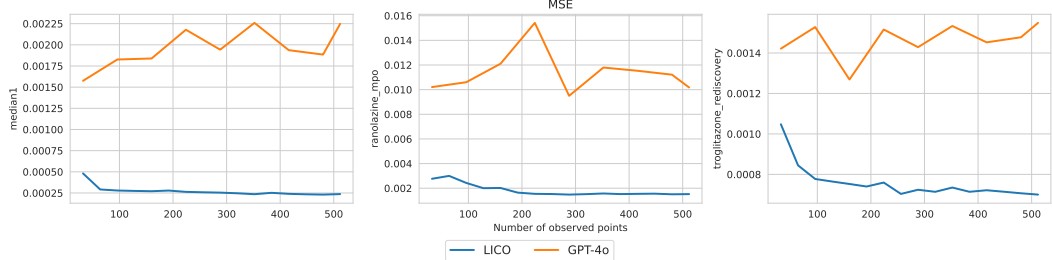

Figure 5: Predictive performance of LICO vs GPT-4o.

Figure 5 compares the predictive performance of LICO with GPT-4o on 3 tasks – `median1`, `ranolazine_mpo`, and `troglitazone_rediscovery`, similar to the paper. We vary the context length from 32 to 512, and for each context length average the mean squared error across 128 target molecules. The result shows that prompting GPT4-o directly in the text space performs poorly, while LICO works much better and its performance improves as we increase the context length.

## C.4 LLM EMBEDDINGS FOR MOLECULAR OPTIMIZATION

This work explored using LLMs as the in-context backbone for surrogate modeling in molecular optimization. Another way to make use of a pretrained LLM is to use the embeddings of an LLM as feature vectors for molecules and train a regressor on top of the embedding space. This section explores this idea with two regression models: a 4-layer MLP and a Gaussian Process (GP), both using the last hidden layer of the frozen LLama-2-7B model as the LLM embeddings. For the MLP baseline, we train the model on the full data buffer for 10 epochs with a batch size of 128 and a learning rate of $1e - 4$, for every 150 molecules collected (every time after each new set of molecules is selected and labeled by the oracle). For the GP baseline, we keep the same hyperparameters as the GP BO method that uses fingerprint features. We compare LICO with these two baselines in the PMO benchmark.

Table 10: Comparison of LICO and other baselines with LLM embeddings on 23 optimization tasks in PMO. A higher score is better. We report the mean and stddev of scores averaged over random seeds. We **bold** the best method for each task.

| Task | LICO | MLP w/ LLM Embedding | GP BO w/ LLM Embedding |
|---|---|---|---|
| albuterol_similarity | $\mathbf{0.885 \pm 0.019}$ | $0.740 \pm 0.149$ | $0.868 \pm 0.175$ |
| amlodipine_mpo | $\mathbf{0.679 \pm 0.027}$ | $0.621 \pm 0.058$ | $0.558 \pm 0.025$ |
| celecoxib_rediscovery | $\mathbf{0.664 \pm 0.122}$ | $0.549 \pm 0.113$ | $0.633 \pm 0.187$ |
| deco_hop | $\mathbf{0.619 \pm 0.015}$ | $0.594 \pm 0.006$ | $0.611 \pm 0.007$ |
| drd2 | $0.928 \pm 0.018$ | $\mathbf{0.931 \pm 0.194}$ | $0.852 \pm 0.263$ |
| fexofenadine_mpo | $\mathbf{0.772 \pm 0.023}$ | $0.734 \pm 0.045$ | $0.709 \pm 0.053$ |
| gsk3b | $\mathbf{0.876 \pm 0.045}$ | $0.793 \pm 0.034$ | $0.845 \pm 0.067$ |
| isomers_c7h8n2o2 | $0.939 \pm 0.022$ | $\mathbf{0.941 \pm 0.116}$ | $0.908 \pm 0.170$ |
| isomers_c9h10n2o2pf2cl | $0.819 \pm 0.039$ | $\mathbf{0.835 \pm 0.111}$ | $0.739 \pm 0.119$ |
| jnk3 | $\mathbf{0.731 \pm 0.037}$ | $0.725 \pm 0.021$ | $0.728 \pm 0.029$ |
| median1 | $0.291 \pm 0.016$ | $0.282 \pm 0.039$ | $\mathbf{0.306 \pm 0.044}$ |
| median2 | $0.280 \pm 0.019$ | $0.223 \pm 0.026$ | $0.280 \pm 0.037$ |
| mestranol_similarity | $0.614 \pm 0.064$ | $\mathbf{0.770 \pm 0.120}$ | $0.612 \pm 0.150$ |
| osimertinib_mpo | $\mathbf{0.820 \pm 0.012}$ | $0.809 \pm 0.011$ | $0.785 \pm 0.012$ |
| perindopril_mpo | $\mathbf{0.557 \pm 0.028}$ | $0.497 \pm 0.024$ | $0.484 \pm 0.034$ |
| qed | $0.936 \pm 0.001$ | $\mathbf{0.946 \pm 0.002}$ | $0.946 \pm 0.002$ |
| ranolazine_mpo | $\mathbf{0.774 \pm 0.008}$ | $0.736 \pm 0.085$ | $0.715 \pm 0.109$ |
| scaffold_hop | $\mathbf{0.547 \pm 0.026}$ | $0.478 \pm 0.012$ | $0.506 \pm 0.010$ |
| sitagliptin_mpo | $\mathbf{0.567 \pm 0.034}$ | $0.544 \pm 0.116$ | $0.429 \pm 0.129$ |
| thiothixene_rediscovery | $0.514 \pm 0.037$ | $0.430 \pm 0.083$ | $\mathbf{0.521 \pm 0.125}$ |
| troglitazone_rediscovery | $\mathbf{0.380 \pm 0.026}$ | $0.299 \pm 0.040$ | $0.370 \pm 0.083$ |
| valsartan_smarts | $0.000 \pm 0.000$ | $0.000 \pm 0.000$ | $0.000 \pm 0.000$ |
| zaleplon_mpo | $\mathbf{0.515 \pm 0.017}$ | $0.500 \pm 0.037$ | $0.483 \pm 0.043$ |
| Sum of scores ($\uparrow$) | **14.708** | 13.975 | 13.887 |

Table 10 shows the superior performance of LICO against the two baselines, achieving the best performance in 14/23 tasks in PMO. In addition to the stronger empirical performance, a significant advantage of LICO is the ability to generalize to any objective function via in-context learning without finetuning.

## D  BROADER IMPACT

Our work studies the application of large language models to black-box optimization, particularly in the domain of molecular optimization. This intersection of machine learning and optimization holds significant promise for advancing our understanding of LLMs' capabilities and limitations, and has significant potential in areas like material science and drug discovery. Our main goal is to enhance machine learning and optimization techniques, but it's also important to consider how these advancements might affect society, such as speeding up the development of new medicines and materials.

## E  COMPUTE RESOURCES

All experiments in this paper are run on a cluster of $4$ A6000 GPUs, each with 49GB of memory.

