# OpenReview forum: "LICO: Large Language Models for In-Context Molecular Optimization"
_ICLR.cc/2025/Conference — ICLR 2025 Poster_

### Official Review · Reviewer_w8JZ · 2024-11-03

**Soundness:** 3
**Presentation:** 3
**Contribution:** 3
**Rating:** 6
**Confidence:** 4

**Summary:**

The authors report an approach to using LLMs with additional embedding layers (similar to FPT) and text-encoding and semi-synthetic training (similar to ExPT) for molecular optimization. They report good performance on the PMO benchmark.

**Strengths:**

- The authors use an established benchmark and perform competitive performance on this benchmark
- Bayesian optimization is one of the most practically useful applications of machine learning in chemistry
- The paper is well-written and easy to follow
- The method is novel (but it is based on combining multiple existing techniques)
- There are useful ablations (e.g., training with various amounts of synthetic data)

**Weaknesses:**

- In some places, the paper seems to indicate that ICL for Bayesian Opt has not been done in chemistry. This, however, is not the case as the following two reports show:
    - https://arxiv.org/abs/2304.05341
    - https://www.researchgate.net/profile/Christoph-Voelker-4/publication/377722231_LLMs_can_Design_Sustainable_Concrete_-a_Systematic_Benchmark_re-submitted_version/links/65b408e934bbff5ba7c85ad8/LLMs-can-Design-Sustainable-Concrete-a-Systematic-Benchmark-re-submitted-version.pdf
- A bit related is the use of LLM-derived embeddings for Bayesian Opt in chemistry. This, for example, has been reported in https://openreview.net/forum?id=A1RVn1m3J3

**Questions:**

- How do you obtain the uncertainties do you directly use the probabilities returned by the model? Are those probabilities well-calibrated?
- The ordering of the examples seems important - what is the impact and how is this taken care of?
- The approach is shown for a causal LM. But it seems that a masked approach (similar to https://www.nature.com/articles/s42256-023-00639-z) might be more effective in learning from the entire sequence

---

> ### Author Response · Authors · 2024-11-20
> **Authors' rebuttal**
>
> We thank the reviewer for their constructive feedback and for appreciating our paper's novelty, competitive performance, and clear writing. We address each specific concern below.
>
> > In some places, the paper seems to indicate that ICL for Bayesian Opt has not been done in chemistry.
>
> We did not intend to say that ICL for Bayesian Opt has not been done in chemistry. **We have added a section in the related work to discuss LLMs for molecular optimization**, including the two papers that the reviewer mentioned. We discuss them briefly below.
> - [1] prompts GPT to perform ICL directly in the text space while LICO embeds each molecule to one token in the feature space of the LLM and performs ICL in this space. This allows LICO to use more context points because one molecule only occupies one token compared to several tokens in the text space. Moreover, [1] obtains uncertainty estimations by using token probabilities from the LLM outputs, while LICO learns a separate prediction head to output the mean and std predictions.
> - [2] similarly prompts GPT in the text space to generate new material designs and/or predict the score/property of a design, while human experts and/or a verifier provide feedback to iteratively improve the generation and prediction.
>
> [1] Ramos, Mayk Caldas, et al. "Bayesian optimization of catalysts with in-context learning." arXiv preprint arXiv:2304.05341 (2023).
>
> [2] Völker, Christoph, et al. "LLMs can Design Sustainable Concrete–a Systematic Benchmark." (2024).
>
> > A bit related is the use of LLM-derived embeddings for Bayesian Opt in chemistry. This, for example, has been reported in https://openreview.net/forum?id=A1RVn1m3J3
>
> Thank you for pointing us to this work. It proposes an orthogonal direction of using LLM embeddings as molecule/chemical representations for Bayesian Opt, while we leverage the pretrained transformer backbone of the LLM to perform in-context learning. **We have cited and discussed this work in the paper**.
>
> > How do you obtain the uncertainties do you directly use the probabilities returned by the model? Are those probabilities well-calibrated?
>
> The new prediction head that we attach to the pretrained LLM outputs both the mean prediction and standard deviation, and **we use the standard deviation to obtain uncertainties**. Figure 2 compares the predictive performance of LICO with a Gaussian Process, and the Root-mean-squared calibration error metric (RMS Cal) shows that **the predicted probabilities are indeed well calibrated**.
>
> > The ordering of the examples seems important - what is the impact and how is this taken care of?
>
> This was also our thought when developing the method. However, some preliminary experiments where we ordered the context points in different ways (random shuffling, sorting according to scores, etc.) indicated that **the performance is very robust to different orderings**. Therefore, in all of the experiments in the paper, we keep the order of context points the same as they appear in the optimization process, i.e., append new context points to the right of the sequence.
>
> > The approach is shown for a causal LM. But it seems that a masked approach (similar to https://www.nature.com/articles/s42256-023-00639-z) might be more effective in learning from the entire sequence
>
> While it’s true that we’re using a causal pretrained LLM, **we note that the model always learns from the entire sequence** when predicting a target point, because we always place the unknown target points at the rightmost position of the sequence. Moreover, **we stick to causal LLMs mainly because the strongest open-source LLMs right now such as Llama are causal** and we want to finetune them in the same way they were pretrained to retain the existing capabilities of the model.
>
> We thank the reviewer again for the constructive feedback and continued support. We believe your suggestion has significantly improved the paper, and **we sincerely hope that the reviewer can take our discussion into account in the final assessment** of the paper.

---

> > ### Author Response · Authors · 2024-11-24
> > **Reminder of our rebuttals**
> >
> > Dear Reviewer w8JZ,
> >
> > We appreciate that you are likely to review multiple other papers; however, as we approach the end of the discussion period (two days left), we would greatly appreciate your feedback on our rebuttals. We believe our rebuttals have addressed the reviewer's two main points: discussion of related work and clarifications on how the model obtains uncertainties and the importance of example orderings. We are happy to address any remaining concerns or questions to improve the paper further.
> >
> > Kind regards,
> > The authors of paper 12932

---

> > > ### Comment · Reviewer_w8JZ · 2024-11-25
> > >
> > > I thank the authors for providing this additional context that will help improve the paper.
> > > My marginal acceptance score has been because the work is well executed - thus a positive score - but it also did not strike me as truly outstanding (which would be required for a strong acceptance).

---

> > > > ### Author Response · Authors · 2024-11-25
> > > > **Thank you**
> > > >
> > > > We thank the reviewer for the constructive feedback and continued support. Your questions and suggestions have greatly improved our paper.

---

### Official Review · Reviewer_8R1V · 2024-11-03

**Soundness:** 3
**Presentation:** 3
**Contribution:** 2
**Rating:** 5
**Confidence:** 3

**Summary:**

This work proposes a semi-synthetic training approach for LLM-based surrogate modeling, with a specific focus on molecular optimization. By integrating a pretrained language model with embedding and prediction layers and training on both "intrinsic" and synthetic data, it shows promises of outperforming gold-standard Gaussian process regressors. As a result, the molecular optimization algorithm coupled with this LLM-based surrogate shows better performances than presented baselines, ranging from RL algorithms to GFN etc. The authors also presented analysis on different model sizes and training approach with varying data mixture.

**Strengths:**

-The overall approach shows good efficacy, as shown in the benchmark scores. The combo of training data and modeling recipe is interesting and novel for surrogate modeling, as far as I know (this needs cross-confirmation).

-The scaling law analysis is interesting, indicating the power of scaling up the model size for better molecule optimization outcome.

-The ablation study and result analysis is helpful: the analysis of surrogate modeling accuracy, vs. the GPR baseline, confirms the efficacy of proposed LLM surrogate modeling approach. The discussion of the limitation of PMO benchmark numbers are also helpful and shows scientific rigor.

**Weaknesses:**

-The authors selectively show the numbers with a different sampling budget (1k instead of 10k in the original PMO setting) with a reason. Can they also present the numbers with different sampling budget in the supplementary information? That will confirm the generalization of the proposed approach.

-The ablation and baseline should be more comprehensive: there are several concurrent works for LLM for molecular optimization[1,2], the authors should also add them as a baseline, if applicable, and thereafter discuss the efficacy of the proposed methods. For example, the MolLEO framework [1] (https://github.com/zoom-wang112358/MOLLEO) claims that they achieve superior performances than baselines on PMO as well, how does it compare with LICO?

-Two more straightforward baselines I can come up with are: (1) drop the input embedding layers, simply extract the text embedding from prompts and train an MLP layer for the surrogate. (2) drop both the embedding and prediction layers, use the pretrained model to do in context learning only. This is similar to the LLAMBO work [3] that is cited in the paper. The authors should justify why they think the proposed approach is the most promising here.

-The description of the experimental details is a bit lacking: e.g. how the embedding and prediction layers are integrated into the existing LLM backbone? Please also provide code and pretrained models so that the reviewers can reproduce the results.

-The language models tested in this work, such as llama2 and qwen1.5, are slightly outdated. The authors should also add numbers on llama3/3.1, Qwen2 to further confirm their conclusions.

-The related works section on LLM for molecular optimization is missing.

[1] Efficient Evolutionary Search Over Chemical Space with Large Language Models, arXiv:2406.16976 .
[2] ChatGPT-powered Conversational Drug Editing Using Retrieval and Domain Feedback, ICLR 2024.
[3] https://github.com/tennisonliu/LLAMBO; Large Language Models to Enhance Bayesian Optimization, ICLR 2024.

**Questions:**

1. Can the authors argue why the additional input embedding layers are necessary here? considering that the LLM always comes with an existing embedding layers.

2. How the embedding and prediction layers are integrated into the existing LLM backbone?

3. Maybe I missed this, but what language representation is used in LICO?

4. What's the rationale behind using Tanimoto kernels for GPR synthetic data generation? Have the authors tried other types of kernels, and how do they perform?

**Details Of Ethics Concerns:**

N.A.

---

> ### Author Response · Authors · 2024-11-22
> **Authors' Rebuttal (1/3)**
>
> We thank the reviewer for the constructive feedback and for recognizing the novelty and efficacy of our proposed method. We address each specific concern below.
>
> > The authors selectively show the numbers with a different sampling budget (1k instead of 10k in the original PMO setting)
>
> **We ran LICO with 10K budget during the rebuttal and updated the paper with the new results**. We also named PMO with 1K budget PMO-1K to clearly distinguish the two settings. LICO outperforms the PMO baselines but slightly underperforms Genetic GFN and Augmented Memory. **We note that it was highly non-trivial to scale an in-context learning method like LICO to 10K context examples**. To do this, we adopted Llama-2-7B-32K-Instruct, the long-context checkpoint of Llama-2 provided by TogetherAI (https://www.together.ai/blog/llama-2-7b-32k-instruct), and utilized the Liger Kernel (https://github.com/linkedin/Liger-Kernel) with gradient checkpointing for finetuning and inferencing with long context. We hope the reviewer will appreciate our efforts in providing this result.
>
> We also wanted to emphasize that **we performed hyperparameter tuning with grid search for each of the baselines on the first 5 tasks in PMO-1K**, then used the optimal hyperparameters for the remaining tasks. The table below specifies our grid search and the optimal set of hyperparameters for each method, which differs from what was reported in the original PMO paper. We did our best to make sure the baselines achieved their best performance in this benchmark. We have included the hyperparameter tuning details in the updated submission. We hope the reviewer will recognize our efforts in ensuring a fair and honest comparison.
>
> | Method           | Grid search                                      | Optimal hyperparameters                         |
> |------------------|--------------------------------------------------|-----------------------------------------------|
> | **GP BO**        | population_size ∈ {50, 100, 150, …, 350}         | population_size = 50                           |
> |                  | offspring_size ∈ {50, 100, 150}                  | offspring_size = 100                           |
> |                  | kept_offspring_size ∈ {5, 10, 15, …, 50}         | kept_offspring_size = 15                       |
> | **Graph GA**     | population_size ∈ {50, 100, 150, …, 350}         | population_size = 50                           |
> |                  | offspring_size ∈ {50, 100, 150}                  | offspring_size = 100                           |
> | **REINVENT**     | batch_size ∈ {4, 8, 16, 32, 64}                  | batch_size = 16                                |
> |                  | experience_replay ∈ {4, 8, 16, 24, 32}           | experience_replay = 24                         |
> | **Genetic GFN**  | learning_rate ∈ {0.0001, 0.0005}                 | learning_rate = 0.0001                         |
> |                  | batch_size ∈ {4, 8, 16, 32, 64}                  | batch_size = 8                                 |
> |                  | Num_keep ∈ {128, 256, 512, 1024}                 | num_keep = 128                                 |
> |                  | offspring_size ∈ {2, 4, 8}                       | offspring_size = 2                             |
> |                  | ga_generations ∈ {1, 2}                          | ga_generations = 1                             |
> | **Augmented Memory** | batch_size ∈ {4, 8, 16, 32, 64}               | batch_size = 32                                |
> |                  | replay_buffer_size ∈ {50, 100, 150}              | replay_buffer_size = 100                       |
>
> > Different sampling budgets to confirm the generalization of the proposed approach
>
> We believe generalization should not depend on the sampling budget but on how many tasks a model can generalize to. In the paper, we show that a single LICO model can generalize to and solve 23 different optimization tasks in the PMO benchmark, which strongly indicates the generalization of our method.

---

> ### Author Response · Authors · 2024-11-22
> **Authors' Rebuttal (2/3)**
>
> > The ablation and baseline should be more comprehensive: there are several concurrent works for LLM for molecular optimization, the authors should also add them as a baseline, if applicable
>
> We agree that MOLLEO and ChatDrug are interesting and relevant works and we have updated the paper to discuss them. **However, they are completely different approaches and are not comparable**:
> - MOLLEO proposed to incorporate LLMs to perform crossover and mutation operations in a standard graph genetic algorithm. As shown in Appendix E of their paper, the performance of MOLLEO largely depends on the alignment of the prompt with the model's knowledge, and the optimal prompt varies between tasks. In contrast, LICO is a general-purpose model that can optimize any objective after training without any finetuning or ad-hoc optimization.
> - ChatDrug was proposed for drug editing. ChatDrug requires a specific prompt design for each task, and requires the task/objective to be described in natural language. In contrast, LICO does not require such domain expertise, and one LICO model can optimize any molecular objective without changing any component. Moreover, the ReDF in ChatDrug requires a retrieval database of molecules that is used to inject domain feedback into the LLM. To perform the injection, the method also assumes it is cheap and easy to evaluate the validity of the molecules in this database for the editing task. These assumptions do not hold in optimization tasks where we do not have a fixed database, and there is not a clear notion of validity to retrieve from such a database for knowledge injection.
>
> > Two more straightforward baselines are: (1) drop the input embedding layers, simply extract the text embedding from prompts and train an MLP layer for the surrogate. (2) drop both the embedding and prediction layers, use the pretrained model to do in context learning only.
>
> We thank the reviewer for this suggestion. We indeed considered these alternatives before coming up with LICO. **To demonstrate (2), we conducted an experiment where we prompted GPT-4o to perform in-context property prediction** in the text space. The figure at https://imgur.com/a/VosqY6o compares the predictive performance of GPT-4o with LICO on 3 tasks – median1, ranolazine_mpo, and troglitazone_rediscovery, similar to the paper. We vary the context length from 32 to 512, and for each context length average the mean squared error across 128 target molecules. The result shows that prompting GPT4-o directly in the text space performs poorly, while LICO works much better and the performance improves as we increase the context length.
>
> In addition to the empirical results, **there are three strong reasons why LICO is better than (1) and (2)**. First, since we train separate embedding and prediction layers, LICO can be easily adapted to any new, even non-textual domain beyond molecules, while using the pretrained text embedding cannot. Moreover, training a separate embedding layer that embeds each molecule as a single token reduces the sequence length significantly. If we represent each molecule by its SMILE string, and assume each string takes 10 tokens on average, then this approach results in 10 times longer sequences, and hence can use 10 times fewer in-context examples, compared to LICO. Last but not least, using a separate prediction head allows LICO to quantify uncertainties which is essential to the optimization algorithm. We have motivated these design choices in lines 211-215 in the main text.
>
> > The description of the experimental details is a bit lacking: e.g. how the embedding and prediction layers are integrated into the existing LLM backbone?
>
> Figure 1 in the paper depicts how we integrate the new embedding and prediction layers into the pretrained LLM backbone. The input sequence is formatted as (task prompt, “x”, x_1, “y”, y_1, “x”, x_2, “y”, y_2, …, “x”, x_n, “y”, y_n), in which x_i is the fingerprint (a 2048-dimensional vector) representation of molecule i, and y_i is the scalar score of molecule i. Given this sequence, we embed the task prompt as well as “x” and “y” using the pretrained text embedding layer of the LLM, while embedding x_i and y_i using their respective MLP embedding layers to the same dimension of the LLM. This produces a sequence of hidden vectors with the same dimension, and we feed this sequence through the transformer layers of the LLM. Finally, the prediction layer takes the feature vector of the last transformer layer and outputs prediction y^_i, which consists of the mean and standard deviation for each molecule x_i.

---

> > ### Author Response · Authors · 2024-11-22
> > **Authors' Rebuttal (3/3)**
> >
> > > The language models tested in this work, such as llama2 and qwen1.5, are slightly outdated.
> >
> > LLM is a fast-moving field and there are stronger and better LLMs every few months. Llama-2, Phi-2, and Qwen-1.5 are still some of the most popular open-source LLMs people use nowadays. Moreover, **this does not affect any of our contributions or conclusions**: we developed a very general method that can be applied to any existing pretrained LLM, and Figure 3 already shows a strong signal that the more powerful the base LLM is, the better LICO performs.
> >
> > > The related works section on LLM for molecular optimization is missing.
> >
> > **We have added a section in the related work to discuss LLMs for molecular optimization**, including Chemlactica/Chemma, DrugAssist, MOLGEN, MOLLEO, and ChatDrug.
> >
> > > Can the authors argue why the additional input embedding layers are necessary here?
> >
> > Please see our answer above.
> >
> > > How the embedding and prediction layers are integrated into the existing LLM backbone?
> >
> > Please see our answer above.
> >
> > > Maybe I missed this, but what language representation is used in LICO?
> >
> > Do you mean the language we used for the task prompt? We used English which is the dominant language used to pretrain the LLMs.
> >
> > > What's the rationale behind using Tanimoto kernels for GPR synthetic data generation? Have the authors tried other types of kernels, and how do they perform?
> >
> > Unlike RBF kernels which assume continuous Euclidean space, **Tanimoto kernels naturally handle the discrete structure of fingerprints and are specifically designed to measure molecular similarity in cheminformatics**. The baseline GP-BO also uses Tanimoto kernels. If one wants to use another kernel like RBF, they should first encode the molecule space to a continuous space, for example by using a pretrained model, and then apply RBF kernels in this space. This is an orthogonal direction to our work.
> >
> > We thank the reviewer again for the constructive feedback and continued support. We believe our discussion has significantly improved and clarified the paper, and **sincerely hope that the reviewer can take our discussion into account** in the final assessment of the paper.

---

> > > ### Author Response · Authors · 2024-11-24
> > > **Reminder of our rebuttals**
> > >
> > > Dear Reviewer 8R1V,
> > >
> > > We appreciate that you are likely to review multiple other papers; however, as we approach the end of the discussion period (two days left), we would greatly appreciate your feedback on our rebuttals. We believe our rebuttals have addressed the reviewer's four main points: 1) benchmarking on the full PMO with 10k oracle calls, 2) discussion of related work on LLM for molecular optimization, 3) experiments with alternative baselines, and 4) clarifications on how embedding and prediction layers are integrated into the LLM backbone. We spent significant time and effort on the additional results and analysis and we are happy to address any remaining concerns or questions to improve the paper further.
> > >
> > > Kind regards,
> > > The authors of paper 12932

---

> ### Comment · Reviewer_8R1V · 2024-11-25
>
> To clarify, I was asking for which type of molecular language representation, x, was used in the LLM. But now it becomes clear to me, that you are using the 2048-dimension vector as x there (also appears in line 350). Is that ECFP/Morgan fingerprints? I am not sure if I missed the details somewhere. Can you also point me to the actual location in your paper that explain the fingerprint details? right now it's unclear what the 2048-dimension vector is, but I assume it's ECFP. Please correct me if I am wrong.
>
> Overall, thank you for your detailed responses on your thoughts of the MOLLEO and other baselines, and running new experiments for 10k-optimization budget, and adding GPT4-o in-context learning as a baseline.
>
> However, several of my major concerns still exist: primarily of the soundness of the designed LLM approach here, as well as the effectiveness compared with the state-of-the-art methods.
>
> > We believe generalization should not depend on the sampling budget but on how many tasks a model can generalize to. In the paper, we show that a single LICO model can generalize to and solve 23 different optimization tasks in the PMO benchmark, which strongly indicates the generalization of our method.
>
> I totally agree with your point that generalization to multiple tasks matters a lot, but I don't think testing on sampling budget is meaningless. For example, right now when extending to 10k budget as the default setting of PMO, the advantage of LICO shrinks than other methods. Furthermore, the PMO results challenge the effectiveness of the method that appears in PMO-1k. (I don't object that your method is good overall, but the trend does change.)
>
> > We agree that MOLLEO and ChatDrug are interesting and relevant works and we have updated the paper to discuss them. However, they are completely different approaches and are not comparable:
>
> ChatDrug may be less related but in terms of MOLLEO, although it's different approach, I don't think it's not comparable if our goal is to push the frontier of PMO (MOLLEO is tested on exactly the same benchmark). Let's take a step back, if it's not comparable since it's a completely different approach, what's the point of comparing any new methods with REINVENT, GFN?
>
> If you do compare with MOLLEO (and/or other more recent methods) and show the advantages of LICO, I believe it will be more convincing.
>
> >  To demonstrate (2), we conducted an experiment where we prompted GPT-4o to perform in-context property prediction in the text space.
>
> Thank you very much for conducting this additional experiment. Can you please add the experimentation details and settings into your paper? as well as the results.
>
> > In addition to the empirical results, there are three strong reasons why LICO is better than (1) and (2). [...]
>
> For (1), what if we use a SMILES representation of a molecule, extract the embedding from the LLM, and then add a prediction head for the target properties? This is a simple and scalable baseline. To keep it simple, you can freeze the LLM weights and only fine-tune the prediction head. I personally think this is a must-have baseline in your work.
>
> > [..] The input sequence is formatted as (task prompt, “x”, x_1, “y”, y_1, “x”, x_2, “y”, y_2, …, “x”, x_n, “y”, y_n), in which x_i is the fingerprint (a 2048-dimensional vector) representation of molecule i, and y_i is the scalar score of molecule i. Given this sequence, [...]
>
> Sorry I missed this information in line 359-360. Thanks for clarifying this. Your answer actually reminds me of a more fundamental question: Why do we want to combine the molecular feature vector into the LLM architecture, is that due to the flexibility of adding the task-specific prompt ahead of it? It's not very clear to me why we can benefit from pretrained LLM if the molecule representation is based on chemistry-informed feature vectors. Have you tried training random forests with these features directly for the surrogating modeling, is that bad?
>
> I look forward to your thoughts on these.

---

> > ### Author Response · Authors · 2024-11-26
> >
> > We thank the reviewer for the constructive feedback and suggestions. We have added the GPT-4o experiment and the additional experiments the reviewer suggested to the updated submission. We address the remaining questions and concerns below.
> >
> > > Is that ECFP/Morgan fingerprints?
> >
> > We used Morgen fingerprints as the molecule representation, similar to the GP BO and several other baselines in PMO.
> >
> > > If you do compare with MOLLEO (and/or other more recent methods) and show the advantages of LICO, I believe it will be more convincing.
> >
> > **We added MOLLEO as a baseline to the updated submission and evaluated it on all 23 tasks for both PMO and PMO-1K**. We used BioT5 as the backbone for MOLLEO because it is open-source and performed well in the original paper. In both evaluation settings, MOLLEO slightly underperforms LICO in terms of the sum score across tasks.
> >
> > We also note that MOLLEO has significant advantages over LICO and other methods. MOLLEO used BioT5 to generate valid molecules, which has been finetuned extensively on molecules, protein, and molecule-related text data, while LICO leveraged a general LLM like Llama. Moreover, MOLLEO prompts the LLM with a detailed textual description of the task such as “Your job is to generate a SELFIES molecule that looks more like a drug", which possibly results in data contamination, since the finetuning data may have included similar tasks. On the other hand, we use LICO as a black-box surrogate model, where the model makes predictions based purely on in-context learning of the mapping between molecule fingerprints and their corresponding scores.
> >
> > > For (1), what if we use a SMILES representation of a molecule, extract the embedding from the LLM, and then add a prediction head for the target properties?
> >
> > We thank the reviewer for this suggestion. **We included this method in the updated submission** with two regression models: a 4-layer MLP and a Gaussian Process (GP), both using the last hidden layer of the frozen LLama-2-7B model as the LLM embeddings. For the MLP baseline, we train the model on the full data buffer for 10 epochs with a batch size of 128 and a learning rate of 1e-4, for every 150 molecules collected (every time after each new set of molecules is selected and labeled by the oracle). For the GP baseline, we keep the same hyperparameters as the GP BO method that uses fingerprint features.
> >
> > In the PMO benchmark, LICO achieves a superior performance against the two baselines, having the best performance in 14/23 tasks in PMO. In addition to the stronger empirical performance, a significant advantage of LICO is the ability to generalize to any objective function via in-context learning without finetuning, whereas models like MLP require training one network for each task.
> >
> > > Why do we want to combine the molecular feature vector into the LLM architecture? It's not very clear to me why we can benefit from pretrained LLM if the molecule representation is based on chemistry-informed feature vectors.
> >
> > **We can benefit from the general in-context learning capabilities of the pretrained LLM**, i.e., the ability to extract the relationship between x (molecule fingerprint) and y (score) in the context points to make predictions for unseen x (molecule fingerprint). We discussed this in detail in lines 209-226. Table 5 in the paper empirically demonstrates this, where we compare LICO with two backbones, one is a pretrained LLM (Llama-2-7B), and the other is a randomly initialized LLM with the same size as the pretrained LLM. The result shows the superior performance of the pretrained LLM, indicating the ability to transfer the in-context learning capabilities of a pretrained LLM to a new domain such as molecules.
> >
> > **Moreover, we are not the first to discover that we can use the pretrained LLM backbone for a completely different domain by simply replacing the embedding layer and the prediction head**. Frozen Pretrained Transformers (FPT) was one of the first papers studying this, and we discussed LICO with FPT in lines 230-239.
> >
> > > Have you tried training random forests with these features directly for the surrogating modeling, is that bad?
> >
> > **The GP BO baseline we included in the paper uses the exact same features as LICO, i.e., 2-radius 2048-bit Morgan molecular fingerprints**. GP BO is a strong baseline in this paper as well as the original PMO paper, but LICO outperforms GP BO in both PMO-1K and PMO. The superior performance of LICO once again shows that we leverage a pretrained LLM backbone as a strong in-context learning engine. We had a paragraph that discussed LICO vs GP BO in lines 419-431 in the paper.
> >
> > We thank the reviewer again for your suggestions, which helped improve the paper significantly. We spent a great amount of time and effort in providing additional experiments in a short period, which we believe have addressed most major concerns. We sincerely hope the reviewer will consider our discussions in the final assessment of the paper.

---

> ### Author Response · Authors · 2024-11-30
> **Reminder of our rebuttals**
>
> Dear Reviewer 8R1V,
>
> We thank you for your constructive feedback and suggestions. It's been a few days since we posted our rebuttals and we sincerely hope the reviewer has had time to read it. Per the reviewer's request, **we have included two more baselines: 1) MLP/GP with LLM embeddings, and 2) MOLLEO**, which we believe have resolved the two major concerns previously raised: soundness of LICO and its effectiveness compared to state-of-the-art methods.
>
> For completeness, **we have also trained random forests for surrogate modeling to compare with LICO** as the reviewer suggested. The figure at https://imgur.com/a/D4aKJHx shows the mean-squared-error predictive performance of LICO vs a 100-tree RF across the first 15 objective functions in PMO with varying numbers of observations. LICO clearly outperforms RF on 12/15 objectives.
>
> As we approach the end of the discussion period, we would greatly appreciate your feedback on our rebuttals. We are happy to address any remaining concerns or questions to improve the paper further. Thank you so much!
>
> Kind regards,
> The authors of paper 12932

---

> > ### Author Response · Authors · 2024-12-01
> > **Reminder of our rebuttals**
> >
> > Dear Reviewer 8R1V,
> >
> > It's been a few days since we posted our rebuttals and we sincerely hope the reviewer has had time to read it. We appreciate that you are busy and likely to review multiple other papers; however, as we approach the end of the discussion period, your feedback on our rebuttals is critically important to us. We believe our rebuttals have addressed the reviewer's main concerns. We are happy to address any remaining problems or questions to improve the paper further.
> >
> > Kind regards,
> > The authors of paper 12932

---

> > > ### Author Response · Authors · 2024-12-02
> > > **Reminder of our rebuttals**
> > >
> > > Dear Reviewer 8R1V,
> > >
> > > It's been almost a week since we posted our rebuttals and we hope the reviewer has had time to read it. **As today is the last day the reviewer can respond to us, we sincerely hope the reviewer will provide feedback on our rebuttals soon because it is critically important to our work.** We believe our rebuttals have addressed the reviewer's main concerns. We are happy to address any remaining problems or questions to improve the paper further.
> > >
> > > Kind regards, The authors of paper 12932

---

### Official Review · Reviewer_7dsL · 2024-11-04

**Soundness:** 3
**Presentation:** 3
**Contribution:** 4
**Rating:** 8
**Confidence:** 4

**Summary:**

This paper introduces LICO, a versatile model that enhances LLMs for black-box optimization, specifically in the molecular domain. LICO overcomes limitations related to domain-specific data scarcity and complex problem expression. It is trained to perform in-context predictions across diverse functions and, post-training, efficiently generalizes to new molecule properties through simple prompting. LICO achieves state-of-the-art PMO molecular optimization benchmark results, demonstrating its efficacy in complex scientific applications.

**Strengths:**

1. The paper studies an important task of adapting LLMs for molecular optimization tasks, which has not been studied extensively.

2. The paper presents a novel approach by integrating LLMs with specialized layers to address black-box optimization problems in the molecular domain.

3. The model achieves strong performance on the challenging PMO benchmark.

**Weaknesses:**

1. While this paper demonstrates the strong performance of LLMs, the analysis of their specific benefits remains limited. It would be valuable to understand which particular characteristics of LLMs contribute to the success of this molecular optimization task. For instance, how do different LLM architectures or configurations impact performance? Would domain-adaptive training on chemistry corpora further enhance results? Expanding on these points with additional explanation would strengthen the understanding of LLMs' effectiveness in this context.

2. The study offers a limited exploration of prompt formats. Further investigation into how different prompt structures might influence model performance would be beneficial (e.g. Prompts that include more domain-specific chemistry terminology/Prompts that frame the task in different ways (e.g. as a prediction task vs. an optimization task/Testing different ways of structuring the input-output pairs within the prompt).

3. It would be better to discuss several recent works for using LLMs for molecule optimization: a) DrugAssist: A Large Language Model for Molecule Optimization; b) Domain-Agnostic Molecular Generation with Self-feedback; c) A Sober Look at LLMs for Material Discovery: Are They Actually Good for Bayesian Optimization Over Molecules?

**Questions:**

See above.

---

> ### Author Response · Authors · 2024-11-20
> **Authors' Rebuttal**
>
> We thank the reviewer for their constructive feedback and for recognizing the importance, novelty, and strong performance of our work. We address each specific concern below.
>
> > It would be valuable to understand which particular characteristics of LLMs contribute to the success of this molecular optimization task.
>
> We believe **the general in-context capabilities of the pretrained LLM are what contribute to its strong performance in a new domain like molecules**. In Section 5.2.3, we compared a pretrained LLM with a model of the same size (7B) but trained from scratch, and the results showed that the pretrained LLM significantly outperforms the scratch model. Another piece of evidence is Figure 3, which demonstrates that larger models with stronger capabilities also perform better in molecular optimization. These results indicate that LICO successfully transfers general in-context learning capabilities of pretrained LLMs in the text domain to a completely new domain such as molecules.
>
> > Would domain-adaptive training on chemistry corpora further enhance results?
>
> We thank the reviewer for bringing up this very interesting point. To answer this question, we compare the performance of LICO with two different base LLMs: T5-base [1] and Nach0-base [2], which finetunes T5-base on molecule corpora. Due to time constraints, we did not perform optimization with these new models, but compared their predictive performance instead. The figure at https://imgur.com/a/v4ih18n summarizes the results. There are two interesting observations from this figure.
> - Nach0 works significantly better than plain T5, **confirming the hypothesis that proper finetuning of a language model on molecule data helps boost its in-context property prediction in LICO**.
> - Llama-2 works better than Nach0. As we explained in the paper, because we perform in-context learning in the embedding space of the language model, we rely on the general pattern-matching capability of the model, i.e., the ability to extract the relationship between embeddings of x and y from examples. From this perspective, it is not surprising that Llama-2 works better than a T5-based model, since it is much larger and has been pretrained with a lot more data, leading to a superior pattern-matching capability. One more benefit of using general LLMs like Llama is that they are domain-agnostic, which means we can finetune them for other non-molecule domains as well. **We have included this result in the updated version of the paper**.
>
> [1] Raffel, Colin, et al. "Exploring the limits of transfer learning with a unified text-to-text transformer." Journal of machine learning research 21.140 (2020): 1-67.
>
> [2] Livne, Micha, et al. "nach0: Multimodal natural and chemical languages foundation model." Chemical Science 15.22 (2024): 8380-8389.
>
> > The study offers a limited exploration of prompt formats. Further investigation into how different prompt structures might influence model performance would be beneficial
>
> Table 2 compares LICO with different language instructions, and it shows that a more structured (but simple) prompt boosts performance. **We intentionally kept the prompt format simple** because 1) we want to develop a general method that is independent of the prompt, and can leverage any existing LLMs without changing the prompt to suit the LLM, and 2) it is not the main focus of our work and the current simple prompt already shows the strong performance of LICO.
>
> > It would be better to discuss several recent works for using LLMs for molecule optimization
>
> We thank the reviewer for pointing us to these related works. **We have cited and discussed them in the updated submission**. We discuss them briefly below.
> - DrugAssist creates the MolOpt-Instructions dataset that contains pairs of molecules and their actual property values to finetune a pretrained LLM, while LICO only uses the unlabeled molecular data. Due to unsupervised finetuning, LICO can generalize to any objective function after training, while DrugAssist can only optimize what it was finetuned on.
> - MOLGEN was an LLM explicitly pretrained and prefix-tuned on molecular corpora to synthesize valid molecules, whereas LICO leverages generic LLMs like Llama for molecular optimization. MOLGEN and LICO are two orthogonal but complementary directions.
> - A Sober Look at LLMs for Material Discovery paper studies two ways to use LLMs for molecular Bayesian optimization: use a pretrained LLM as a feature extractor, or finetune an LLM to serve as the surrogate. Unlike LICO, they did not study synthetic pretraining or in-context learning. Therefore, they had to finetune the LLM for each specific task, while LICO generalizes to arbitrary objective functions after training.
>
> We thank the reviewer again for the constructive feedback and continued support. Your suggestion has significantly improved the paper, and **we sincerely hope that the reviewer will consider our discussion in the final assessment of the paper**.

---

> > ### Author Response · Authors · 2024-11-24
> > **Reminder of our rebuttals**
> >
> > Dear Reviewer 7dsL,
> >
> > We appreciate that you are likely to review multiple other papers; however, as we approach the end of the discussion period (two days left), we would greatly appreciate your feedback on our rebuttals. We believe our rebuttals have addressed the reviewer's three main points: 1)  investigation of domain-adaptive training on chemistry corpora, 2) prompt explorations, and 3) discussion of related work. We are happy to address any remaining concerns or questions to improve the paper further.
> >
> > Kind regards,
> > The authors of paper 12932

---

> > > ### Comment · Reviewer_7dsL · 2024-11-25
> > >
> > > Thanks for your comments. It helps to address my concerns. I also appreciate the results of additional studies on the PMO benchmark. I have updated my score accordingly.

---

> > > > ### Author Response · Authors · 2024-11-25
> > > > **Thank you**
> > > >
> > > > We thank the reviewer for the constructive feedback and continued support. We believe your questions and suggestions have significantly improved our work.

---

### Official Review · Reviewer_MdPk · 2024-11-04

**Soundness:** 3
**Presentation:** 4
**Contribution:** 4
**Rating:** 6
**Confidence:** 5

**Summary:**

The paper focuses on one of the most promising directions of modern LLMs: LLM-enhanced optimization algorithms. It suggests a method to extend arbitrary pretrained LLMs with a couple of layers and train them to perform in-context learning for arbitrary functions. The idea is tested on molecular property prediction tasks, and the paper claims SOTA results on the famous PMO benchmark.

**Strengths:**

* the "domain adaption" trick to convert the general purpose text-based LLMs into domain-specific in-context learners using synthetic data.
* detailed ablation studies. I really liked the analysis of the effect of the ratio of "intrinsic" and "synthetic" datasets. It gives an intuition on how to design synthetic datasets for other optimization tasks in other domains.
* the "scaling law" chart (Fig. 3) is a good indicator of the scaling abilities of the proposed approach. Unfortunately there is a diversity of underlying models which makes the claims less significant. Hopefully there will be many sizes of similarly pretrained LLMs at some point (e.g. Llama 4 1B, 3B, 8B, etc.).

**Weaknesses:**

The main weakness is that the SOTA claim on PMO is misleading. **The results reported in this paper are not really PMO.** There are two major differences.

a) PMO has 23 tasks, not 21. *jnk3* and *gsk3* are missing.

b) PMO uses 10K budget of oracle calls (as mentioned by the authors).

While a) does not make the comparison to prior art unfair, b) is critical. The main advantage of the PMO paper was that the authors performed a large-scale hyperparameter search for every method they tried, and even discovered hyperparameter values (e.g. for REINVENT) that were not covered even by the authors of the methods. So, all methods from PMO, and the subsequent methods like Genetic GFN have their hyperparameters tuned for the 10K budget.

I agree with the authors that 1K budget can be more interesting, as 10K might feel saturated, but that's a different benchmark. I would suggest to name it something like **PMO-1K**, and then properly tune the hyperparameters of the baselines. I understand it's hard to do this in the review discussion period.

** Details of the optimization algorithm**
It took me a few days to understand that the sentence "At each iteration t, we generate a set of candidates" in Section 4.3 does not mean that the candidates are generated without the LLM. As seen in the Appendix, the authors actually used a manually designed genetic algorithm for generating the candidates, and the LLM is only used for scoring them. This is a critical component of the algorithm and has to be presented well in the main part of the paper.

**Three other papers that could be cited and discussed:**
1. Optformer [1] is the earliest transformer to the best of my knowledge that used in-context learning for an optimization task. It did not use an initialization from a large pretrained model, and the data used there is not synthetic. Still, the concept is very close.
2. Another recent approach that produced good scores on PMO is from the Chemlactica/Chemma models [2]. It has the genetic algorithm idea, very similar to the one described in the Appendix A.3. Chemlactica's scores on PMO are still a bit unfair, as it uses a lot more molecules in the pretraining phase (way beyond ZINC250k).
3. MOLLEO [3] is another evolutionary algorithm that wraps an LLM. It has a few evaluations on

A minor aspect that could be considered in the future iterations: use more realistic oracles, like molecular docking. Check [2] and [4] for new benchmarks.

[1] Chen, Yutian, et al. "Towards learning universal hyperparameter optimizers with transformers." Advances in Neural Information Processing Systems 35 (2022): 32053-32068.
[2] Guevorguian, Philipp, et al. "Small Molecule Optimization with Large Language Models." arXiv preprint arXiv:2407.18897 (2024).
[3] Wang, Haorui, et al. "Efficient evolutionary search over chemical space with large language models." arXiv preprint arXiv:2406.16976 (2024).
[4] Guo, Jeff, and Philippe Schwaller. "Saturn: Sample-efficient Generative Molecular Design using Memory Manipulation." arXiv preprint arXiv:2405.17066 (2024).

**Questions:**

**Suggestion:**
I am willing to increase my rating if the authors report results on the *original* PMO-10K, which will ensure fair comparison with the prior work. Even if the results are not SOTA anymore, the paper can still be accepted, as the methods of the paper are interesting on their own.

The paper can have an additional table for PMO-1K, with either tuned baselines, or with a note that the baselines are not carefully tuned.

---

> ### Author Response · Authors · 2024-11-21
> **Authors' Rebuttal (1/2)**
>
> We thank the reviewer for their constructive feedback, and for appreciating the novelty and detailed experiments of our work. We address each specific concern of the reviewer below:
>
> > PMO uses 10K budget of oracle calls (as mentioned by the authors).
>
> We thank the reviewer for this suggestion. **We ran LICO with 10K budget during the rebuttal and updated the paper with the new results**. We also named PMO with 1K budget PMO-1K to clearly distinguish the two settings. LICO outperforms the PMO baselines but slightly underperforms Genetic GFN and Augmented Memory. **We note that it was highly non-trivial to scale an in-context learning method like LICO to 10K context examples**. To do this, we adopted Llama-2-7B-32K-Instruct, the long-context checkpoint of Llama-2 provided by TogetherAI (https://www.together.ai/blog/llama-2-7b-32k-instruct), and utilized the Liger Kernel (https://github.com/linkedin/Liger-Kernel) with gradient checkpointing for finetuning and inferencing with long context. We hope the reviewer will appreciate our efforts in providing this result.
>
> > All methods from PMO, and the subsequent methods like Genetic GFN have their hyperparameters tuned for the 10K budget.
>
> We note that **we performed hyperparameter tuning with grid search for each of the baselines on the first 5 tasks in PMO-1K**, then used the optimal hyperparameters for the remaining tasks. The table below specifies our grid search and the optimal set of hyperparameters for each method, which differs from what was reported in the original PMO paper. We did our best to make sure the baselines achieved their best performance in this benchmark. We have included the hyperparameter tuning details in the updated submission. We hope the reviewer will recognize our efforts in ensuring a fair and honest comparison.
>
> | Method           | Grid search                                      | Optimal hyperparameters                         |
> |------------------|--------------------------------------------------|-----------------------------------------------|
> | **GP BO**        | population_size ∈ {50, 100, 150, …, 350}         | population_size = 50                           |
> |                  | offspring_size ∈ {50, 100, 150}                  | offspring_size = 100                           |
> |                  | kept_offspring_size ∈ {5, 10, 15, …, 50}         | kept_offspring_size = 15                       |
> | **Graph GA**     | population_size ∈ {50, 100, 150, …, 350}         | population_size = 50                           |
> |                  | offspring_size ∈ {50, 100, 150}                  | offspring_size = 100                           |
> | **REINVENT**     | batch_size ∈ {4, 8, 16, 32, 64}                  | batch_size = 16                                |
> |                  | experience_replay ∈ {4, 8, 16, 24, 32}           | experience_replay = 24                         |
> | **Genetic GFN**  | learning_rate ∈ {0.0001, 0.0005}                 | learning_rate = 0.0001                         |
> |                  | batch_size ∈ {4, 8, 16, 32, 64}                  | batch_size = 8                                 |
> |                  | Num_keep ∈ {128, 256, 512, 1024}                 | num_keep = 128                                 |
> |                  | offspring_size ∈ {2, 4, 8}                       | offspring_size = 2                             |
> |                  | ga_generations ∈ {1, 2}                          | ga_generations = 1                             |
> | **Augmented Memory** | batch_size ∈ {4, 8, 16, 32, 64}               | batch_size = 32                                |
> |                  | replay_buffer_size ∈ {50, 100, 150}              | replay_buffer_size = 100                       |
>
> > PMO has 23 tasks, not 21. jnk3 and gsk3 are missing.
>
> We excluded GSK3B and JNK3 in the original submission because we had a version mismatch problem with scikit-learn when trying to load the machine learning oracles. We fixed it and **included the results for GSK3B and JNK3 for both PMO and PMO-1K in the updated submission**.

---

> ### Author Response · Authors · 2024-11-21
> **Authors' Rebuttal (2/2)**
>
> > Three other papers that could be cited and discussed
>
> We thank the reviewer for pointing us to these relevant works. We have cited and discussed them in the updated submission. We briefly discuss them below.
> - As mentioned by the reviewer, Optformer does not leverage a pretrained LLM. Moreover, Optformer operates in the raw text space, which leads to a more limited context length than LICO and does not work for non-textual domains.
> - Chemlactica/Chemma is trained on a massive molecule-related corpus and works as a genetic algorithm to sample new molecules, whereas LICO is used to score generated molecules from a genetic algorithm. Similarly to Optformer, Chemlactica/Chemma also works directly in the text space. Chemlactica/Chemma and LICO are orthogonal but complementary approaches for using LLMs for molecular optimization.
> - MOLLEO proposed to prompt a pretrained LLM to perform crossover and mutation operations in a standard graph genetic algorithm. As shown in Appendix E, MOLLEO’s performance largely depends on the alignment of the prompt with the model's knowledge, and the optimal prompt varies between tasks. In contrast, LICO is a more general-purpose method that can optimize any objective after training without any ad-hoc optimization.
>
> > Details of the optimization algorithm
>
> We agree with the reviewer that missing this detail can cause confusion. We have updated the paper to move the detail in A.3 to the main text.
>
> > A minor aspect that could be considered in the future iterations: use more realistic oracles, like molecular docking.
>
> We agree that testing LICO on more realistic oracles is an interesting and important direction. However, we believe the diversity of tasks in PMO has showcased the effectiveness and generalization of our method. We will continue adopting and adapting LICO to more oracles in future work.
>
> We thank the reviewer again for the constructive feedback and continued support. We believe your suggestion has significantly improved the paper, and **sincerely hope that the reviewer can take our discussion into account** in the final assessment of the paper.

---

> > ### Author Response · Authors · 2024-11-24
> > **Reminder of our rebuttals**
> >
> > Dear Reviewer MdPk,
> >
> > We appreciate that you are likely to review multiple other papers; however, as we approach the end of the discussion period (two days left), we would greatly appreciate your feedback on our rebuttals. We believe our rebuttals have addressed the reviewer's two main points: benchmarking on the full PMO and discussion of related work. We are happy to address any remaining concerns or questions to improve the paper further.
> >
> > Kind regards,
> > The authors of paper 12932

---

> > > ### Author Response · Authors · 2024-11-25
> > > **Reminder of our rebuttals**
> > >
> > > Dear Reviewer MdPk,
> > >
> > > We appreciate that you are likely to review multiple other papers; however, as we approach the end of the discussion period (there is only one day left), we would greatly appreciate your feedback on our rebuttals. We are happy to address any remaining concerns or questions to improve the paper further. Thank you so much!
> > >
> > > Kind regards,
> > > The authors of paper 12932

---

> > > > ### Author Response · Authors · 2024-11-26
> > > > **Reminder of our rebuttals**
> > > >
> > > > Dear Reviewer MdPk,
> > > >
> > > > We appreciate that you are likely to review multiple other papers; however, as we approach the end of the discussion period, we would greatly appreciate your feedback on our rebuttals. We are happy to address any remaining concerns or questions to improve the paper further. Thank you so much!
> > > >
> > > > Kind regards, The authors of paper 12932

---

> > > > > ### Author Response · Authors · 2024-11-28
> > > > > **Reminder of our rebuttals**
> > > > >
> > > > > Dear Reviewer MdPk,
> > > > >
> > > > > We appreciate that you are likely to review multiple other papers; however, as we approach the end of the discussion period, we would greatly appreciate your feedback on our rebuttals. We are happy to address any remaining concerns or questions to improve the paper further. Thank you so much!
> > > > >
> > > > > Kind regards,
> > > > > The authors of paper 12932

---

> > > > > > ### Author Response · Authors · 2024-11-30
> > > > > > **Reminder of our rebuttals**
> > > > > >
> > > > > > Dear Reviewer MdPk,
> > > > > >
> > > > > > It's been more than a week since we posted our rebuttal. We appreciate that you are likely to review multiple other papers; however, as we approach the end of the discussion period, we would greatly appreciate your feedback on our rebuttals. We are happy to address any remaining concerns or questions to improve the paper further. Thank you so much!
> > > > > >
> > > > > > Kind regards,
> > > > > > The authors of paper 12932

---

> > > > > > > ### Author Response · Authors · 2024-11-30
> > > > > > > **Reminder of our rebuttals**
> > > > > > >
> > > > > > > Dear Reviewer MdPk,
> > > > > > >
> > > > > > > It's been more than a week since we posted our rebuttal. We appreciate that you are busy and likely to review multiple other papers; however, as we approach the end of the discussion period, your feedback on our rebuttals is critically important to us. We believe our rebuttals have addressed the reviewer's two main points: benchmarking on the full PMO and discussion of related work. We are happy to address any remaining concerns or questions to improve the paper further.
> > > > > > >
> > > > > > > Kind regards, The authors of paper 12932

---

> > > > > > > > ### Author Response · Authors · 2024-12-01
> > > > > > > > **Reminder of our rebuttals**
> > > > > > > >
> > > > > > > > Dear Reviewer MdPk,
> > > > > > > >
> > > > > > > > It's been more than a week since we posted our rebuttal. We appreciate that you are busy and likely to review multiple other papers; however, as we approach the end of the discussion period, your feedback on our rebuttals is critically important to us. We believe our rebuttals have addressed the reviewer's two main points: benchmarking on the full PMO and discussion of related work. We are happy to address any remaining concerns or questions to improve the paper further.
> > > > > > > >
> > > > > > > > Kind regards,
> > > > > > > > The authors of paper 12932

---

> > > > > > > > > ### Author Response · Authors · 2024-12-02
> > > > > > > > > **Reminder of our rebuttals**
> > > > > > > > >
> > > > > > > > > Dear Reviewer MdPk,
> > > > > > > > >
> > > > > > > > > It's been more than 10 days since we posted our rebuttal. **As today is the last day the reviewer can respond to us, we sincerely hope the reviewer will provide feedback on our rebuttals soon because it is critically important to our work**. We believe our rebuttals have addressed the reviewer's two main points: benchmarking on the full PMO and discussion of related work. We are happy to address any remaining concerns or questions to improve the paper further.
> > > > > > > > >
> > > > > > > > > Kind regards,
> > > > > > > > > The authors of paper 12932

---

### Meta-Review · Area_Chair_39cE · 2024-12-22

**Metareview:**

The paper introduces LICO, a general-purpose model that extends arbitrary base Large Language Models (LLMs) for black-box optimization, with a particular focus on the molecular domain. The authors equip the language model with separate embedding and prediction layers, training it to perform in-context predictions on a diverse set of functions. LICO is shown to perform competitively on the PMO benchmark. The paper claims that LICO can generalize to unseen molecule properties via in-context prompting, leveraging the pattern-matching capabilities of pretrained LLMs.

Strengths:
* Novelty/Usefulness: The paper presents a novel approach by integrating LLMs with specialized layers for black-box optimization in molecular domains
* Detailed Experiments and Analysis: The authors provide extensive experiments and ablation studies, demonstrating the effectiveness of LICO on the PMO benchmark.
* Promising Scaling Laws: The scaling law analysis suggests the method should continue to improve as the underlying model grows.

Weaknesses:
* **Unusual Benchmarking**. The experiments focus (and did so exclusively before the rebuttal) on a non-standard use of the PMO benchmark, which unexplicably excluded two datasets and used a much-smaller-than-usual sampling budget (1K calls instead of 10K). Only after multiple reviewers raised concerns about the fairness of comparisons (most other methods have parameters tailored to the 10K setting) did the authors include the usual 10K budget results, where their method’s advantage is much less clear or completely disappears in some datasets.
* **Reproducibility**: The description of experimental details is somewhat lacking, making it difficult to reproduce the results.
* **Few Explaining Insights**: The paper provides little in the way of explanation for what aspects (architecture, pretraining, prompting) of the LLMs contribute to the success of this molecular optimization task.
*  **Missing Baselines**: The paper does not adequately compare LICO with recent methods, limiting the understanding of its relative performance. Some of these were adding to the related work section during the rebuttal phase, but were not included as baselines in the experiments.

Overall, despite the potentially promising method proposed in this paper, there were various issues around benchmarking fairness, comparison, and selective result presentation that unfortunately cast a shadow over its contribution. Additionally, it has been brought to our attention that this paper had already been called out for non-standard use of the PMO benchmark during the NeurIPS 2024 rebuttal phase, where it was ultimately rejected. The same issues were brought up there: excluding two of the 23 tasks, and using a much more limited number of oracle calls than standard practice. After the reviewers raised those issues during that discussion phase, the authors included the missing datasets in the comparison and pledged to disclose the decision to focus on a smaller-than-usual oracle budget in the revised version of the paper. Unfortunately, they did not: in the version they originally submitted here to ICLR, which had a deadline more than a month after the NeurIPS rebuttal phase, the authors again left out those two tasks, and doubled down on the unusual 1K calls budget without fully disclosing its atypicality. The issue came up again in one of the reviews and was addressed anew in the rebuttal.

In our deliberation with SACs, we decided to give the authors the benefit of the doubt here. Overall, the decision to accept/reject this paper was very difficult due to the above reasons. In the end, we recommend acceptance mostly because of the potential interest and impact that this paper will have in the community, which given the novelty and results—even if partially clouded in uncertainty— is likely to be moderate to strong.

If the paper is accepted, the authors need to make sure that the revised results and the accompanying discussion of the oracle budget are indeed included in the camera-ready version. As a general advice for the authors: **Modifying evaluation conditions for standardized benchmarks should be carefully and transparently justified a priori, always**.

**Additional Comments On Reviewer Discussion:**

During the rebuttal period, several key points were raised by the reviewers:
1. Benchmarking and Task Exclusion: Reviewers MdPk and 8R1V highlighted the exclusion of two tasks from the PMO benchmark and the use of a 1K budget. The authors addressed this by including the missing tasks and providing results for the 10K budget, but concerns about the initial omissions persisted.
2. Experimental Details and Reproducibility: Reviewers 8R1V and w8JZ requested more detailed experimental descriptions and additional baselines. The authors provided some additional experiments and clarified certain aspects, but the overall reproducibility remained a concern.
3. Comparison with Recent Methods: Reviewers 8R1V and 7dsL suggested including comparisons with recent methods like MOLLEO and ChatDrug. The authors added some comparisons, but the effectiveness of LICO relative to these methods was still questioned.
4. Generalization and Performance: Reviewers appreciated LICO’s generalization capabilities but noted its diminished performance at higher budgets. The authors argued that LICO’s strength lies in low-budget scenarios, but this did not fully address concerns about its scalability.

In weighing these points in my decision, the seemingly promising results of the paper were heavily affected by the persistent issues with benchmarking integrity, reproducibility, and comparative evaluation, despite the authors’ efforts to address some of the concerns. Those alone could be grounds for rejection. Ultimately, I am willing to give the paper the benefit of the doubt given its potential impact in the community.

---

### Decision · Program_Chairs · 2025-01-22

Accept (Poster)